# EEGMamba: Bidirectional State Space Model with Mixture of Experts for EEG Multi-task Classification

## Abstract

In recent years, with the development of deep learning, electroencephalogram (EEG) classification networks have achieved certain progress. Transformer-based models can perform well in capturing long-term dependencies in EEG signals. However, their quadratic computational complexity poses a substantial computational challenge. Moreover, most EEG classification models are only suitable for single tasks and struggle with generalization across different tasks, particularly when faced with variations in signal length and channel count. In this paper, we introduce EEGMamba, the first universal EEG classification network to truly implement multi-task learning for EEG applications. EEGMamba seamlessly integrates the Spatio-Temporal-Adaptive (ST-Adaptive) module, bidirectional Mamba, and Mixture of Experts (MoE) into a unified framework. The proposed ST-Adaptive module performs unified feature extraction on EEG signals of different lengths and channel counts through spatial-adaptive convolution and incorporates a class token to achieve temporal-adaptability. Moreover, we design a bidirectional Mamba particularly suitable for EEG signals for further feature extraction, balancing high accuracy, fast inference speed, and efficient memory-usage in processing long EEG signals. To enhance the processing of EEG data across multiple tasks, we introduce task-aware MoE with a universal expert, effectively capturing both differences and commonalities among EEG data from different tasks. We evaluate our model on eight publicly available EEG datasets, and the experimental results demonstrate its superior performance in four types of tasks: seizure detection, emotion recognition, sleep stage classification, and motor imagery. The code is set to be released soon.

## 1 Introduction

Electroencephalogram (EEG) is a technique of recording brain activity using electrophysiological indicators, which captures the electrical wave changes during brain activity. EEG can be utilized to detect various human physiological activities such as seizure detection, emotion recognition, motor imagery, sleep stage classification, and other physiological related task (Shoeibi et al., 2021; Jafari et al., 2023; Altaheri et al., 2023; Sri et al., 2022).

In recent years, with the development of deep learning, EEG classification models based on deep learning have been widely used (Chen et al., 2022). Among them, models based on Convolutional Neural Networks (CNNs) and Transformers are the most representative, each with their own strengths and weaknesses. CNN-based EEG classification networks have the advantage of faster training and inference speeds, and they perform well on short EEG signals. However, due to the lack of global sequence modeling ability, their performance on long EEG signals cannot be guaranteed (Sakhavi et al., 2018; Thuwajit et al., 2021; Schirrmeister et al., 2017). In contrast, Transformer-based EEG classification networks have good capability of global sequence modeling, achieving excellent performance on both short and long EEG signals. Nevertheless, as the length of the EEG signals increases, the computational complexity of the model increases quadratically, significantly raising the training and inference costs (Dai et al., 2023; Xie et al., 2022; Wang et al., 2022).

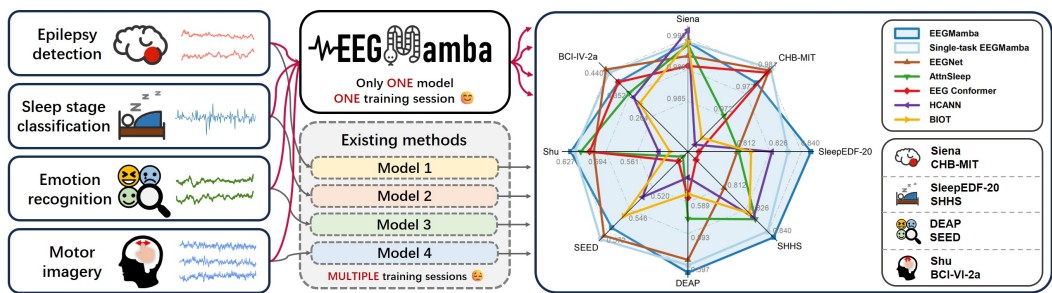

Figure 1: Our proposed EEGMamba can simultaneously process EEG signals from multiple tasks including epilepsy detection, sleep stage classification, emotion recognition, and motor imagery. It achieves state-of-the-art (SOTA) performance on the majority of datasets.

Recently, State Space Models (SSM) with selection mechanism and efficient hardware-aware design, such as Mamba (Gu & Dao, 2023), have shown great potential in long sequence modeling. By utilizing selective state space model, it effectively captures the relationships between tokens in a sequence, addressing the limitation of CNNs in modeling long sequences. Moreover, it exhibits linear computational complexity, which outperforms the quadratic complexity of Transformers and provides a strong backbone network for training EEG classification models on long EEG signals.

Single-task learning (STL) is the most commonly used paradigm in current EEG classification models (O'Shea et al., 2020; Phan et al., 2022; Algarni et al., 2022; Autthasan et al., 2021), where each task is learned independently given a set of learning tasks. For example, EEGNet (Lawhern et al., 2018) has been validated on four different tasks but can only address one type of task in a single training session. In contrast, multi-task learning (MTL) trains models by simultaneously learning all tasks and sharing representations across related ones, which enabling the model to learn more robust and universal representations for multiple tasks compared to single-task model (Choo et al., 2023). Therefore, designing a classification network capable of handling multi-task EEG data simultaneously might be a promising approach.

Few previous studies have employed multi-task classification in EEG, and they all have certain limitations (Prodhan et al., 2022; Li et al., 2022). For instance, (Li et al., 2022) achieved simultaneous classification tasks across four emotion evaluation metrics using the same dataset, but its multi-task classification ability is limited to handling multiple labels within a single dataset. The lack of models capable of performing EEG classification across multiple different datasets may be due to the highly challenging problems.

One of the significant obstacles for multi-task EEG classification is that different EEG data have varying numbers of channels and signal lengths, which makes it difficult for networks to adapt during a single training. For example, MaskSleepNet (Zhu et al., 2023) can classify EEG signals with different numbers of channels by manually setting the channel parameter, but it uses a fixed-parameter Multi-scale CNN that can only process EEG signals with limited input lengths. While EEG ConvNet (Schirrmeister et al., 2017) is designed with a structure capable of adapting to arbitrary signal lengths, it still requires manual setting in different trainings. Therefore, enabling the model to adapt to different signal lengths and channel counts represents a significant challenge.

On the other hand, EEG data from different tasks show both differences and commonalities, making it challenging for models without specialized multi-task processing module to capture these relationships, ultimately leading to interference between tasks. Mixture of Experts (MoE) is a deep learning model with sparse gate-controlled architecture, consisting of a group of expert models and a gating network (Jacobs et al., 1991; Shazeer et al., 2016; Xue et al., 2024). The gating network can dynamically select experts to specifically process input data, enabling the network to accurately distinguish and better process multi-task data, thus reducing interference between tasks. Therefore, using MoE to achieve EEG multi-task classification might be a feasible solution.

In general, existing EEG classification models mainly face two challenges. First, these models find it difficult to balance high accuracy, fast inference speed, and efficient memory-usage when dealing with long EEG signals. Second, they often struggle to handle different EEG classification tasks and demonstrate poor generality.

To address the aforementioned two issues, we propose EEGMamba, which utilizes bidirectional Mamba suitable for EEG signals, as well as a Spatio-Temporal-Adaptive (ST-Adaptive) module and task-aware MoE for targeted processing of multi-task EEG classification. Our model enhances Mamba by employing bidirectional modeling to capture the relationships between tokens in a one-dimensional temporal sequence, achieving high accuracy and fast inference speed. Additionally, we propose an ST-Adaptive module that uses spatial-adaptive convolution to process EEG signals of varying channel numbers and a class token to achieve temporal adaptability without any additional processing. To efficiently capture differences and commonalities between EEG data from different tasks, we design a task-aware gating network that accurately directs different EEG task tokens to specific experts for processing, while also employing a universal EEG expert to exploit commonalities among different EEG tasks. In summary, our contributions are as follows:

- **Bidirectional Mamba Design for EEG Signals.** We introduce bidirectional Mamba specifically for EEG signals, achieving the balance between fast inference speed, efficient memory-usage and excellent global perception ability.

- **First Implementation of Multi-task Learning in EEG application.** EEGMamba is the first model to truly implement multi-task learning for EEG classification, enabling a more integrated and effective analysis of complex brain signal data.

- **ST-Adaptive Module for Flexible EEG Processing.** We propose an ST-Adaptive module that can automatically adapt to EEG signals of different lengths and channels, allowing for simultaneous processing in single training session.

- **Task-aware MoE for EEG Data.** We design Task-aware MoE with a universal expert, achieving the capture of both differences and commonalities between EEG data from different tasks.

## 2 METHOD

EEGMamba primarily consists of the ST-Adaptive module, BiMamba, and task-aware MoE. The ST-Adaptive module processes EEG signals of arbitrary lengths and channel numbers through spatial-adaptive convolution, tokenize layer, and temporal-adaptation based on the class token. The features extracted by the ST-Adaptive module are then processed by multiple BiMamba blocks and task-aware MoE modules. The BiMamba block allows the model to effectively capture long-term dependencies in EEG signals, while the task-aware MoE enables targeted processing of EEG features for different tasks. Finally, a task-aware classifier provides the classification results. The overall model architecture is illustrated in Figure 2.

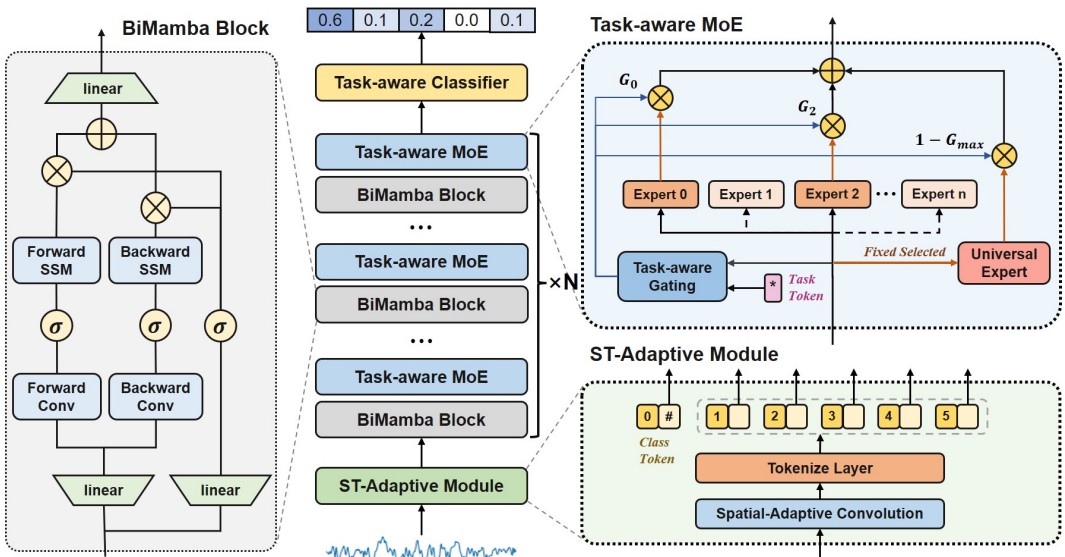

Figure 2: Overall structure of EEGMamba. The model consists of ST-Adaptive module, Bidirectional Mamba (BiMamba) blocks and Task-aware MoE modules.

## 2.1 PRELIMINARY WORK

Mamba is inspired by continuous state space equations. For continuous input $x(t) \in \mathbb{R}$ in the time domain, the corresponding output $y(t) \in \mathbb{R}$ is determined by the current hidden state $h(t)$ and input $x(t)$ at time $t$, as shown in Equation (1). Here, $A \in \mathbb{R}^{N \times N}$ is the state matrix, $B \in \mathbb{R}^{N \times 1}$ is related to the system's hidden state, and $C \in \mathbb{R}^{1 \times N}$ is a parameter associated with the input and output.

$$h'(t) = Ax(t) + Bh(t) \tag{1}$$
$$y(t) = Ch(t)$$

Mamba discretizes the continuous time $t$ into discrete time, transforming the continuous state space equations into discrete state space equations. Specifically, by introducing a time-scale parameter $\Delta$, $A$ and $B$ are transformed into discrete time parameters $\bar{A}$ and $\bar{B}$ respectively. The zero-order hold (ZOH) technique is used as the transformation rule, as shown in Equation (2).

$$\bar{A} = exp(\Delta A) \tag{2}$$
$$\bar{B} = (\Delta A)^{-1}(exp(\Delta A) - I)\Delta B$$

In practice, following the approach of (Gu & Dao, 2023), we approximate $\bar{B}$ using a first-order Taylor expansion, as show in Equation (3):

$$\bar{B} = (\Delta A)^{-1}(exp(\Delta A) - I)\Delta B \approx \Delta B \tag{3}$$

Finally, the discretized form of the continuous state space equation is shown in Equation (4).

$$h_t = \bar{A}h_{t-1} + \bar{B}x_t \tag{4}$$
$$y_t = Ch_t$$

Based on the mentioned discrete state-space equations, Mamba further introduces data dependency into the model parameters, enabling the model to selectively propagate or forget information based on the sequential input tokens. In addition, it utilizes a parallel scanning algorithm to accelerate the equation solving process.

## 2.2 ST-ADAPTIVE MODULE

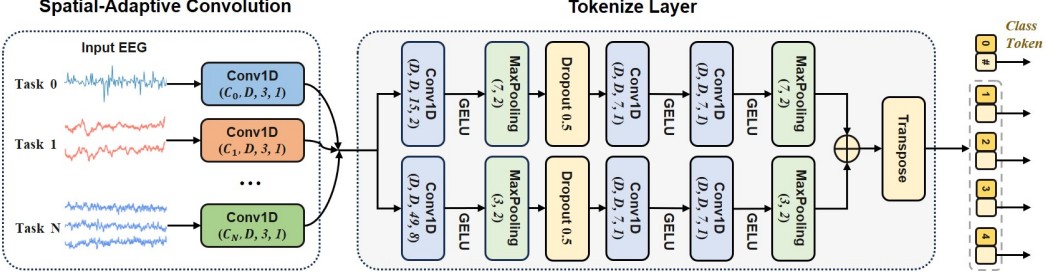

Figure 3: Overall structure of ST-Adaptive module.

EEG signals from different datasets often have different lengths and channel numbers. To address this issue, we propose a Spatio-Temporal-Adaptive module that transforms input signals of arbitrary lengths and channel numbers into uniform feature dimension, as shown in Figure 3.

To handle the inconsistency in the number of input channels, we introduce a spatial-adaptive convolutional module, which standardizes the data to a fixed number of channels. This module consists of a series of 1D-CNN sub-modules, each designed with a uniform output channel count but adaptable to varying input channels. Through this approach, EEG data with different channel numbers are processed uniformly. Let $x \in \mathbb{R}^{B \times C_i \times L_i}$ represent the EEG signals, where $C_i$ denotes the number of EEG channels for the $i$-th task, and $L_i$ is the EEG signal length for the $i$-th task.

$$y_{SA} = CNN_{SA}(x) \in \mathbb{R}^{B \times D \times L_i} \tag{5}$$

As shown in Equation (5), $y_{SA}$ is the result obtained through spatial-adaptive convolution, where the channel dimension is changed from $C_i$ determined by the task $i$ to a unified $D$. Then, $y_{SA}$ is converted into an EEG token sequence through the tokenize layer. In order to better extract features from EEG signals, we design a dual-path structure utilizing a small kernel convolution module $CNN_S$ and a wide convolutional module $CNN_W$. Obtain the small kernel feature token sequence $z_s$ and the wide kernel feature token sequence $z_w$, respectively. Finally, we concatenate them in the time dimension to form the EEG token sequence $T$, as shown in Equation (6).

$$z_s = \mathcal{T}(CNN_s(y_{SA})) \in \mathbb{R}^{B \times N_s \times D}$$

$$z_w = \mathcal{T}(CNN_w(y_{SA})) \in \mathbb{R}^{B \times N_w \times D}$$

$$T = Concat(z_s, z_w, dim = 1) \in \mathbb{R}^{B \times N \times D}$$

(6)

Among them, $\mathcal{T}$ represents the transpose operation, $N_s$, $N_w$, $N$ are the number of EEG small kernel feature tokens, EEG wide kernel feature tokens, and overall EEG tokens, respectively.

Due to the varying lengths of EEG signals, the number of EEG tokens (i.e., the length of the token sequence $T$) obtained from the tokenize layer is inconsistent. To address this issue, we introduce a temporal-adaptive module that incorporates a special class token (Dosovitskiy et al., 2021) for final classification. Specifically, we concatenate this class token with the previously extracted feature token sequence $t_s^1, t_s^2, ..., t_s^{N_s}$ and $t_w^1, t_w^2, ..., t_w^{N_w}$ to obtain the token sequence $T$, as shown in Equation (7).

$$T = [t_{cls}, t_s^1, t_s^2, ..., t_s^{N_s}, t_w^1, t_w^2, ..., t_w^{N_w}] \in \mathbb{R}^{B \times (N+1) \times D}$$

(7)

Then, the input token sequence $T$ is processed through a network (using bidirectional Mamba blocks in this study) to integrate EEG token sequence information into the class token. This approach prevents the network from developing biases towards certain tokens in the EEG feature token sequence $T$ due to variations in input length, thereby achieving temporal adaptability.

## 2.3 BIDIRECTIONAL MAMBA BLOCK FOR EEG SIGNALS

Mamba is designed for Natural Language Processing (NLP), with its output at each moment depends only on the current input and hidden state, without consideration for future time steps. Since NLP is primarily a generative autoregressive task that relies on previous information for judgment, Mamba's single-directional modeling approach is sufficient to complete such tasks. However, EEG classification tasks require simultaneous processing of both preceding and following information, which cannot be learned by single-directional modeling. Therefore, for EEG signals, the original Mamba's single-directional modeling is insufficient.

To address this issue, we design a bidirectional Mamba for one-dimensional temporal signals, which can model the input bidirectionally and more effectively learn the dependencies between time series tokens. We use the features extracted by the ST-Adaptive module as the input for the first bidirectional Mamba block.

---

**Algorithm 1** Bidirectional Mamba Block Process

---

**Input:** token sequence $T_{k-1} \in \mathbb{R}^{B \times (N+1) \times D}$
**Output:** token sequence $T_k \in \mathbb{R}^{B \times (N+1) \times D}$
1: $T_{k-1}^{norm} \leftarrow LayerNorm(T_{k-1})$
2: $X_{k-1} \leftarrow Linear_X(T_{k-1}^{norm}), Z_{k-1} \leftarrow Linear_Z(T_{k-1}^{norm})$
3: $Y_{k-1}^f \leftarrow SSM_f(Conv_f(Transpose(X_{k-1})))$
4: $Y_{k-1}^b \leftarrow Reverse(SSM_b(Conv_b(Reverse(Transpose(X_{k-1})))))$
5: $T_{k-1}' \leftarrow Linear_D(Transpose(Y_{k-1}^f + Y_{k-1}^b) \odot SiLU(Z_{k-1}))$
6: $T_k = T_{k-1}' + T_{k-1}$

---

We denote the input of the bidirectional Mamba block as a sequence $T_{k-1}$ and the output as a sequence $T_k$. First, $T_{k-1}$ is normalized to $T_{k-1}^{norm}$ by layer normalization. Next, it is mapped by $Linear_X$ and $Linear_Z$ to $X_{k-1}$ and $Z_{k-1}$, respectively. Then, $X_{k-1}$ enters parallel forward and

backward sequence modeling modules. The forward module includes forward 1D causal convolution $Conv_f$ and forward SSM module $SSM_f$. Similarly, the backward module includes backward 1D causal convolution $Conv_b$ and backward SSM module $SSM_b$. Then, the results of forward sequence modeling $Y_{k-1}^f$ and backward sequence modeling $Y_{k-1}^b$ are summed with $Z_{k-1}$ through gating and then projected through a linear layer $Linear_D$ to obtain $T_{k-1}'$. Finally, the output sequence $T_k$ is obtained through residual connection. The detailed process is shown in Algorithm 1.

## 2.4 TASK-AWARE MoE WITH UNIVERSAL EXPERT

### 2.4.1 SPARSELY-ACTIVATED MoE

A typical Mixture of Experts (MoE) usually consists of several experts, and each expert is typically represented as a Multi-Layer Perceptron (MLP) whose activation is controlled by a gating network (Shazeer et al., 2016). We define $N_e$ as the number of experts, $E_i$ as the $i$-th expert, and $G$ as the gating network. For each input EEG token sequence $T$, the output $T^*$ of MoE can be expressed as Equation (8):

$$T^* = \sum_{i=1}^{N_e} e_i(T) * E_i(T) \tag{8}$$

$$e_i(T) = SoftMax(Top_k(G(T), k))_i$$

$$\text{Top}_k(V, k)_i = \begin{cases} v_i, & \text{if } v_i \text{ is top } k \text{ value of } V \\ -\infty, & \text{otherwise} \end{cases}$$

### 2.4.2 TASK-AWARE GATING NETWORKS

A gating network calculates gating values based on the input tokens and selects top $k$ experts for activation, typically implemented using a fully connected layer $Linear_{Gate}$. However, this can lead to the problem that only a few experts are trained. To avoid this, we adopted the method from (Shazeer et al., 2016), adding noise to the gating value computation process using a fully connected layer $Linear_{Noise}$, which increases randomness and helps in balancing the load among the experts.

Furthermore, we propose a task-aware gating network which helps improve the accuracy of experts in processing different types of EEG tokens. Specifically, we encode the EEG task into task tokens $t_{task} \in \mathbb{R}^{B \times D}$, then concatenate $t_{task}$ with the EEG token sequence $T$ to obtain $T_{cat}$, which is then sent to the gating network. The gating values calculated in this manner incorporate task information, allowing for better assignment of different tasks to different experts. The working process of the task-aware gating network is shown in Equation (9), where $\epsilon$ represents standard Gaussian noise.

$$T_{cat} = Concat(T, BroadCast(t_{task}), dim = -1) \tag{9}$$

$$G(T, t_{task}) = Linear_{Gate}(T_{cat}) + \epsilon * SoftPlus(Linear_{Noise}(T_{cat}))$$

### 2.4.3 EEG UNIVERSAL EXPERT

EEG signals from different tasks exhibit both differences and commonalities. Only using different experts to process EEG tokens might overlook the connections between tokens from different tasks. Therefore, we design an EEG universal expert that can process EEG tokens from all different tasks and capture their commonalities. To achieve this function, the universal expert is activated for any inputs and not controlled by the gating network's output values.

Overall, our MoE module includes both task experts and a universal expert. Task experts can accurately process EEG tokens from different tasks according to gating values, while universal experts can process all EEG tokens. The output of MoE is the weighted sum of these two types of experts. We adopted a weight design scheme similar to (Gou et al., 2023), as shown in Equation (10). Here, the output weight $\omega$ of the universal expert is determined by the maximum gating value:

$$T^* = \sum_{i=1}^{N_e} e_i(T) * E_i(T) + \omega * E^u(T) \tag{10}$$

$$\omega = 1 - Max(e(T))$$

## 3 EXPERIMENTAL SETUP

### 3.1 DATASET

We evaluate the proposed EEGMamba by using eight datasets from four different tasks, including Siena Scalp EEG Database (Detti et al., 2020), CHB-MIT (Shoeb, 2009), SleepEDF-20 (Kemp et al., 2000), SHHS (Quan et al., 1997), DEAP (Koelstra et al., 2011), SEED (Duan et al., 2013), Shu (Ma et al., 2022), and BCI-IV-2a (Brunner et al., 2008). Table 1 provides an overview of each dataset. For different tasks, the number of classes, the number of channels and the optimal EEG segment length tend to vary depending on the specific task performed. In the experiment, we predefine the number of channels and classes for each EEG dataset.

Table 1: Dataset introduction. '# Sample' refers to the total number of samples used for training and testing after preprocessing steps. More details about the datasets can be found in the appendix D.

| Datasets | Tasks | # Subjects | # Sample | # Classes | # Channels | Rate | Duration |
|---|---|---|---|---|---|---|---|
| Siena | Epilepsy detection | 13 from 14 | 78,958 | 2 | 29 | 512 Hz | 4 seconds |
| CHB-MIT | Epilepsy detection | 23 | 111,678 | 2 | 23 | 256 Hz | 4 seconds |
| SleepEDF-20 | Sleep stage classification | 20 | 33,847 | 5 | 1 | 100 Hz | 30 seconds |
| SHHS | Sleep stage classification | 329 from 6441 | 259,799 | 5 | 1 | 125 Hz | 30 seconds |
| DEAP | Emotion recognition | 32 | 1,040 | 2 | 4 | 128 Hz | 60 seconds |
| SEED | Emotion recognition | 15 | 60,912 | 3 | 62 | 200 Hz | 20 seconds |
| Shu | Motor imagery | 25 | 9,579 | 2 | 32 | 250 Hz | 4 seconds |
| BCI-IV-2a | Motor imagery | 9 | 3,948 | 4 | 22 | 250 Hz | 3 seconds |

### 3.2 IMPLEMENTATION DETAILS

**Data Preprocessing.** We only employ minimal necessary preprocessing. First, we apply a band-pass filter to the EEG signals, retaining components between 0.1 Hz and 50 Hz to remove low-frequency drift and high-frequency noise. Then, we standardize the sampling rate of all EEG signals to 200 Hz. In addition, the public versions of some datasets have undergone some preprocessing. We include a detailed introduction in the Appendix D.

**Data Division.** In all experiments, including the baseline comparison experiments and ablation experiments, we employ five-fold cross-validation grouped by subjects, so that EEG data from the same subject only appear in one fold. Details of the subject division scheme are provided in the Appendix E.3.

**Environments.** The experiments are implemented by Python 3.9.18, PyTorch 2.0.1 + CUDA 12.2 on a Linux server with 256 GB memory. All models are trained on Intel(R) Xeon(R) Gold 6342 CPU and a Nvidia A100 GPU 80G.

Our detailed training strategy, hyperparameter settings, metrics, and baselines are provided in Appendix E.4, E.5, E.6, and F.

## 4 RESULTS AND DISCUSSION

### 4.1 SINGLE-TASK EEGMAMBA PERFORMANCE COMPARISON

The single-task EEGMamba experiment aims to demonstrate the effectiveness of the Mamba-based model. In this experiment, we modify the model by removing MoE modules and redundant spatial-adaptive convolution branches, so the single-task EEGMamba only consists of the essential CNN modules and BiMamba modules. We compare the performance of single-task EEGMamba with previous classification models on eight datasets, as shown in Figure 1. Obviously, single-task EEGMamba outperforms the other non Mamba-based models on the majority of datasets.

We also discuss the memory-usage and inference speed of single-task EEGMamba and Transformer-based models, particularly for long sequences. Figure 4a and Figure 4b show the results for single-channel and multi-channel (here 20 channels) data, respectively. The Transformer-based models in baselines include AttnSleep, EEG Conformer and HCANN. As signal length increases, the memory-usage of Transformer-based models grows quadratically, while single-task EEGMamba grows linearly.

In terms of inference speed, Transformer-based models slow down sharply with longer sequences, while the speed of single-task EEGMamba decreases gently. HCANN performs well on single-channel data due to structural modifications on classical Transformer, but it experiences a significant increase in memory-usage and a notable decrease in inference speed when handling multi-channel data. Overall, single-task EEGMamba comprehensively outperforms Transformer-based models in memory-usage and inference speed.

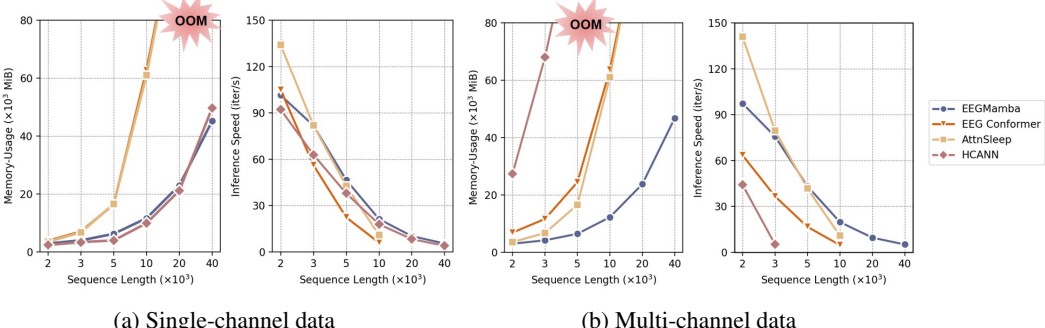

(a) Single-channel data          (b) Multi-channel data

Figure 4: Memory-usage and inference speed of Single-task EEGMamba compared with Transformer-based models. OOM indicates out of memory.

To summarize, compared with the previous classification networks, single-task EEGMamba achieves better performance, lower memory-usage and faster inference speed when dealing with long EEG signals, which roundly demonstrates the feasibility of the Mamba-based model on EEG signals.

## 4.2 EEGMAMBA FOR EEG MULTI-TASK CLASSIFICATION

Table 2, 3, 4 and 5 show the performance of EEGMamba on different datasets compared with several state-of-the-art (SOTA) baselines. EEGMamba ranks among the top three on seven datasets and achieves the best performance on four datasets.

It is worth noting that all classification networks, except EEGMamba, are trained on a single dataset. Single datasets typically have consistency in data distribution, features, and labels, which allows the model to better adapt and optimize for specific patterns of that dataset, thus improving accuracy. Nevertheless, EEGMamba outperforms existing SOTA models across multiple datasets and showed superior overall performance, demonstrating its strong generalization ability to integrate EEG signals from different tasks.

Table 2: Performance of EEGMamba compared with baselines on seizure detection task.

| Methods | Multi-task | Siena | | | CHB-MIT | | |
| --- | --- | --- | --- | --- | --- | --- | --- |
| | | ACC | AUROC | F1 | ACC | AUROC | F1 |
| EEGNet (Lawhern et al., 2018) | ✗ | 0.9886 ± 0.0033 | 0.8828 ± 0.0360 | 0.6905 ± 0.0185 | 0.9814 ± 0.0024 | 0.9064 ± 0.0607 | 0.7690 ± 0.0488 |
| AttnSleep (Eldele et al., 2021) | ✗ | 0.9895 ± 0.0032 | 0.9066 ± 0.0196 | 0.6918 ± 0.0588 | 0.9723 ± 0.0190 | 0.9048 ± 0.0465 | 0.7549 ± 0.0657 |
| EEGConformer (Song et al., 2022) | ✗ | 0.9878 ± 0.0044 | 0.8744 ± 0.0377 | 0.6366 ± 0.0273 | 0.9810 ± 0.0040 | 0.8917 ± 0.0927 | 0.7507 ± 0.0648 |
| BIOT (Yang et al., 2023) | ✗ | 0.9897 ± 0.0043 | 0.8986 ± 0.0223 | **0.7301 ± 0.0550** | 0.9678 ± 0.0284 | 0.8996 ± 0.0831 | 0.7278 ± 0.0886 |
| LaBraM (Jiang et al., 2024) | ✗ | 0.9886 ± 0.0043 | 0.8023 ± 0.0820 | 0.6370 ± 0.0694 | 0.9742 ± 0.0099 | 0.8624 ± 0.0534 | 0.7176 ± 0.0713 |
| HCANN (Ji et al., 2024) | ✗ | **0.9906 ± 0.0026** | **0.9283 ± 0.0208** | 0.6714 ± 0.1115 | 0.9664 ± 0.0227 | 0.9110 ± 0.0572 | 0.7680 ± 0.1203 |
| Single-task EEGMamba | ✗ | 0.9897 ± 0.0053 | 0.9137 ± 0.0105 | 0.7106 ± 0.0326 | **0.9817 ± 0.0036** | 0.9084 ± 0.0437 | 0.7712 ± 0.0600 |
| EEGMamba | ✓ | 0.9897 ± 0.0038 | 0.9082 ± 0.0179 | 0.7070 ± 0.0260 | 0.9789 ± 0.0132 | **0.9126 ± 0.0492** | **0.7964 ± 0.0444** |

**Bold** for the best, red for the second, and underlined for the third.

Table 3: Performance of EEGMamba compared with baselines on sleep stage classification task.

| Methods | Multi-task | SleepEDF-20 | | | SHHS | | |
| --- | --- | --- | --- | --- | --- | --- | --- |
| | | ACC | AUROC | F1 | ACC | AUROC | F1 |
| EEGNet (Lawhern et al., 2018) | ✗ | 0.8165 ± 0.0254 | 0.9464 ± 0.0109 | 0.7322 ± 0.0225 | 0.8174 ± 0.0173 | 0.9351 ± 0.0078 | 0.6663 ± 0.0064 |
| AttnSleep (Eldele et al., 2021) | ✗ | 0.8172 ± 0.0346 | 0.9383 ± 0.0123 | 0.7244 ± 0.0270 | 0.8366 ± 0.0169 | 0.9557 ± 0.0053 | 0.7270 ± 0.0153 |
| EEGConformer (Song et al., 2022) | ✗ | 0.7998 ± 0.0486 | 0.9385 ± 0.0220 | 0.7118 ± 0.0392 | 0.8000 ± 0.0154 | 0.9343 ± 0.0069 | 0.6543 ± 0.0085 |
| BIOT (Yang et al., 2023) | ✗ | 0.8226 ± 0.0387 | 0.9536 ± 0.0147 | 0.7455 ± 0.0315 | 0.8331 ± 0.0152 | 0.9501 ± 0.0103 | 0.7243 ± 0.0287 |
| LaBraM (Jiang et al., 2024) | ✗ | 0.7503 ± 0.0388 | 0.9212 ± 0.0177 | 0.6603 ± 0.0392 | 0.7785 ± 0.0243 | 0.9282 ± 0.0132 | 0.6527 ± 0.0201 |
| HCANN (Ji et al., 2024) | ✗ | 0.8316 ± 0.0396 | 0.9589 ± 0.0129 | 0.7573 ± 0.0387 | 0.8355 ± 0.0167 | 0.9581 ± 0.0077 | 0.7425 ± 0.0117 |
| Single-task EEGMamba | ✗ | 0.8387 ± 0.0399 | 0.9608 ± 0.0116 | 0.7681 ± 0.0359 | 0.8441 ± 0.0163 | 0.9578 ± 0.0074 | 0.7387 ± 0.0155 |
| EEGMamba | ✓ | **0.8486 ± 0.0276** | **0.9636 ± 0.0107** | **0.7738 ± 0.0293** | **0.8478 ± 0.0177** | **0.9587 ± 0.0077** | **0.7433 ± 0.0160** |

**Bold** for the best, red for the second, and underlined for the third.

Table 4: Performance of EEGMamba compared with baselines on emotion recognition task.

| Methods | Multi-task | DEAP | | | SEED | | |
|---|---|---|---|---|---|---|---|
| | | ACC | AUROC | F1 | ACC | AUROC | F1 |
| EEGNet (Lawhern et al., 2018) | ✗ | 0.5979 ± 0.0341 | 0.5906 ± 0.0325 | 0.5624 ± 0.0214 | 0.5739 ± 0.0544 | 0.7448 ± 0.0565 | 0.5561 ± 0.0486 |
| AttnSleep (Eldele et al., 2021) | ✗ | 0.5930 ± 0.0173 | 0.5941 ± 0.0346 | 0.5590 ± 0.0112 | 0.4808 ± 0.0232 | 0.6717 ± 0.0318 | 0.4900 ± 0.0295 |
| EEGConformer (Song et al., 2022) | ✗ | 0.5905 ± 0.0351 | 0.5500 ± 0.0275 | 0.5545 ± 0.0222 | 0.4861 ± 0.0172 | 0.6642 ± 0.0302 | 0.4846 ± 0.0302 |
| BIOT (Yang et al., 2023) | ✗ | 0.5900 ± 0.0165 | 0.5703 ± 0.0283 | 0.5495 ± 0.0310 | 0.5507 ± 0.0591 | 0.7363 ± 0.0666 | 0.5453 ± 0.0700 |
| LaBraM (Jiang et al., 2024) | ✗ | 0.5822 ± 0.0321 | 0.5453 ± 0.0301 | 0.5202 ± 0.0304 | OOM | OOM | OOM |
| HCANN (Ji et al., 2024) | ✗ | 0.5881 ± 0.0226 | 0.5878 ± 0.0350 | 0.5083 ± 0.0484 | 0.5284 ± 0.0282 | 0.7061 ± 0.0589 | 0.5101 ± 0.0361 |
| Single-task EEGMamba | ✗ | 0.5985 ± 0.0247 | 0.5721 ± 0.0184 | 0.5505 ± 0.0157 | **0.5779 ± 0.0584** | **0.7636 ± 0.0514** | **0.5718 ± 0.0580** |
| EEGMamba | ✔ | **0.5994 ± 0.0134** | **0.5957 ± 0.0209** | **0.5628 ± 0.0262** | 0.5646 ± 0.0366 | 0.7538 ± 0.0413 | 0.5583 ± 0.0326 |

**Bold** for the best, red for the second, and underlined for the third.

Table 5: Performance of EEGMamba compared with baselines on motor imagery task.

| Methods | Multi-task | Shu | | | BCI-IV-2a | | |
|---|---|---|---|---|---|---|---|
| | | ACC | AUROC | F1 | ACC | AUROC | F1 |
| EEGNet (Lawhern et al., 2018) | ✗ | 0.5971 ± 0.0454 | 0.6529 ± 0.0708 | 0.6077 ± 0.0538 | **0.4721 ± 0.0570** | **0.7449 ± 0.0591** | **0.4888 ± 0.0683** |
| AttnSleep (Eldele et al., 2021) | ✗ | 0.6105 ± 0.0454 | 0.6464 ± 0.0698 | 0.6061 ± 0.0515 | 0.3807 ± 0.0384 | 0.6376 ± 0.0240 | 0.3747 ± 0.0229 |
| EEGConformer (Song et al., 2022) | ✗ | 0.6014 ± 0.0392 | 0.6418 ± 0.0643 | 0.6064 ± 0.0494 | 0.4228 ± 0.0421 | 0.6856 ± 0.0359 | 0.4136 ± 0.0471 |
| BIOT (Yang et al., 2023) | ✗ | 0.5186 ± 0.0051 | 0.5183 ± 0.0050 | 0.5116 ± 0.0090 | 0.3398 ± 0.0483 | 0.5970 ± 0.0561 | 0.2983 ± 0.0307 |
| LaBraM (Jiang et al., 2024) | ✗ | 0.5368 ± 0.0312 | 0.5426 ± 0.0413 | 0.5343 ± 0.0326 | 0.2879 ± 0.0160 | 0.5333 ± 0.0214 | 0.2804 ± 0.0209 |
| HCANN (Ji et al., 2024) | ✗ | 0.5302 ± 0.0229 | 0.5136 ± 0.0051 | 0.4131 ± 0.0530 | 0.3635 ± 0.0353 | 0.6112 ± 0.0336 | 0.3258 ± 0.0422 |
| Single-task EEGMamba | ✗ | 0.6169 ± 0.0467 | 0.6597 ± 0.0653 | 0.6145 ± 0.0437 | 0.4596 ± 0.0547 | 0.7180 ± 0.0541 | 0.4556 ± 0.0543 |
| EEGMamba | ✔ | **0.6207 ± 0.0505** | **0.6645 ± 0.0681** | **0.6183 ± 0.0525** | 0.4231 ± 0.0522 | 0.6873 ± 0.0542 | 0.4156 ± 0.0545 |

**Bold** for the best, red for the second, and underlined for the third.

Additionally, the multi-task training of EEGMamba provides significant advantages in terms of convenience. First, it is an end-to-end system that does not require separate pre-training and fine-tuning stages, yet offers stronger generalization ability than the pre-trained model. Furthermore, to obtain the corresponding results presented in Table 2 to 5, EEGMamba only needs to be trained once. In contrast, other classification networks require multiple training sessions, each time involving manual adjustments to data length, channel count, and class numbers, making the process much more cumbersome.

## 4.3 VISUALIZATION OF TASK-AWARE MoE IN MULTI-TASK CLASSIFICATION

We explore the role of designed task-aware MoE in practical applications. Since the EEGMamba model contains eight independent MoE modules, we focus our discussion on the last one MoE module as an example. We calculate the activation probability of each expert for different tasks in the task-aware MoE, as shown in Figure 5. The x-axis represents the index of experts, and the y-axis represents their activation probabilities.

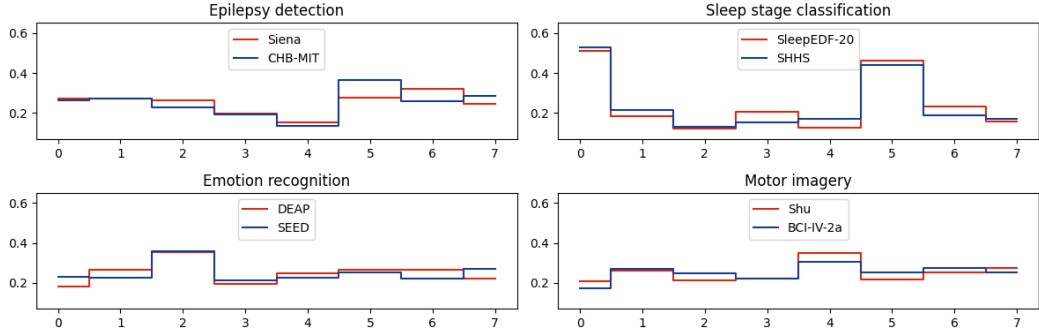

Figure 5: Activation probabilities of MoE experts in the final layer.

When using task-aware MoE, the model exhibits a clear preference for specific experts based on the given task, with different tasks evidently favoring different experts. Specifically, different tasks tend to activate different experts, while data from the same task show similar expert selection probabilities. For instance, experts 5 and 6 are preferred for epilepsy detection, while experts 0 and 5 are favored for sleep stage classification, demonstrating how task-aware MoE enhances flexibility by dynamically adapting to different tasks. This targeted expert selection not only improves task-specific performance

but also maintains efficient processing by bypassing irrelevant experts, thereby reducing unnecessary computational overhead.

### 4.4 ABLATION STUDY

To evaluate the effectiveness of each component in EEGMamba, we conduct ablation experiments on four model variants, including: (i) *Single-directional Mamba*: EEGMamba with Single-directional Mamba; (ii) *EEGMamba w/o MoE*: EEGMamba without the whole MoE module; (iii) *Vanilla MoE*: EEGMamba with the vanilla MoE; (iv) *EEGMamba w/o Task-aware Gating*: EEGMamba without the Task-aware Gating in MoE; (v) *EEGMamba w/o Universal Expert*: EEGMamba without the Universal Expert in MoE.

Figure 6 presents a comparison of ablation experiments on eight datasets across four tasks. EEG-Mamba outperforms other variants on all metrics for all tasks, demonstrating the contribution of each component in our framework. In comparison to the full EEGMamba, the performance of *Single-directional Mamba* shows a significant decline, emphasizing the importance of employing bidirectional Mamba for EEG classification task modeling. Moreover, the performance decline of *EEGMamba w/o MoE* indicates that MoE plays a role in learning the distinctions between different tasks in multi-task classification. In most tasks, the performance of *EEGMamba w/o Task-aware Gating* and *EEGMamba w/o Universal Expert* is similar but slightly lower than the full EEGMamba.

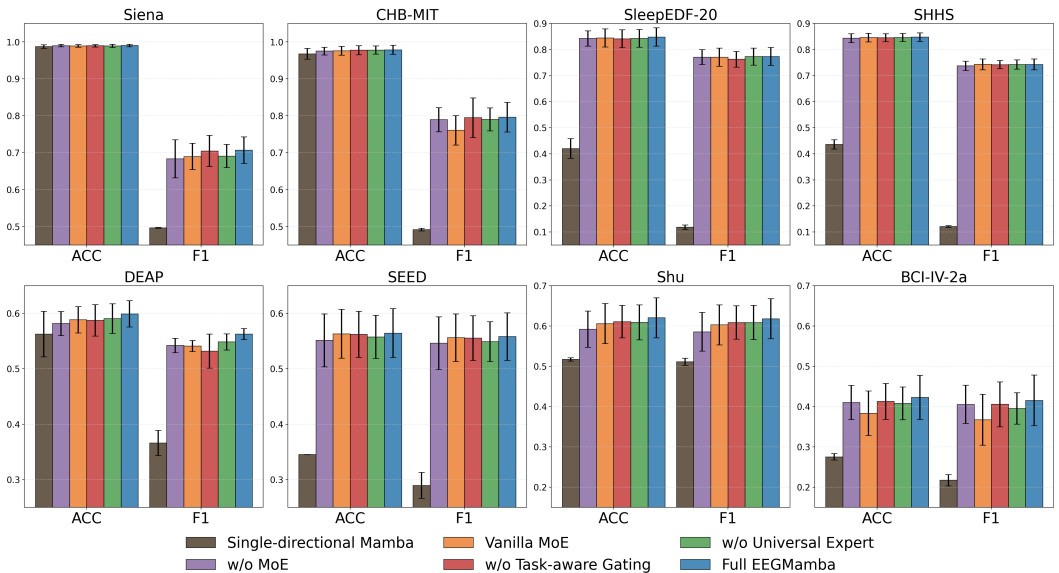

Figure 6: Results of the ablation study on different datasets.

### 5 CONCLUSION

In this paper, we propose EEGMamba, the first model that truly implements multi-task learning for EEG applications. EEGMamba integrates a Spatio-Temporal-Adaptive module to adaptively extract features of EEG data with different lengths and channel counts. We introduce bidirectional Mamba to achieve high accuracy and fast inference speed when processing long-term EEG datasets. Moreover, we design a task-aware Mixture of Experts (MoE) and an EEG universal expert, allowing the model to process multiple tasks simultaneously and better learn the commonalities among EEG signals from different tasks. Our experiments across eight publicly available EEG datasets from four tasks demonstrate the superior performance of our proposed model in multi-task classification scenarios. Our work fills the gap in multi-task classification research within EEG applications, paving the way for future development in this field.

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

# A    RELATED WORKS

## A.1    EEG CLASSIFICATION

The development of deep learning has greatly advanced EEG classification tasks. CNNs are a classic type of neural network with mature applications in EEG classification. (Schirrmeister et al., 2017) proposed a shallow convolutional network with both spatiotemporal convolutional layers to decode task-related information from raw EEG signals. Similarly, (Lawhern et al., 2018) introduced EEGNet, a classic EEG classification network based on depthwise separable convolution, which has demonstrated stable and robust performance in various EEG classification tasks. Recurrent Neural Networks (RNNs) are proposed to capture temporal dependencies in time-series EEG signals. (Supratak et al., 2017) used the RNN architecture for sleep stage classification. (Chen et al., 2020) used CNN and Long Short Term Memory (LSTM) networks for sleep stage classification.

EEG classification networks based on Transformers have also made significant progress. (Eldele et al., 2021) introduced attention mechanisms into EEG classification networks for classifying sleep stages. (Song et al., 2022) proposed EEG Conformer, a EEG classification network based on spatio-temporal convolution and Transformers. EEG Conformer effectively extracts local and global features from EEG signals, and it performs well in tasks such as motor imagery and emotion recognition. HCANN (Ji et al., 2024) combined the multi-head mechanism with CNN to extract complementary representation information from multiple subspaces, making it more suitable for EEG signals. It has achieved state-of-the-art performance on three datasets from different tasks.

In recent years, there has been notable progress in pre-trained EEG classification networks. (Yang et al., 2023) proposed BIOT, a generic biosignal learning model that employs a tokenization module and was evaluated on several EEG, ECG, and human sensory datasets. (Yi et al., 2023) proposed a pre-training framework named MMM, which follows the approach of Masked Auto-Encoder (MAE) for pre-training and employs a multi-stage pre-training strategy to enhance the robustness of the representations.

## A.2    STATE SPACE MODEL

A state space model is a mathematical model that represents a physical system as a set of input, output, and state variables related by a first-order differential equation. (Gu et al., 2021) proposed the Structured State-Space Sequence Model (S4) to model long-term dependencies. (Smith et al., 2022) introduced a new S5 layer by incorporating Multiple Input Multiple Output (MIMO) SSM and efficient parallel scanning within the S4 layer. (Fu et al., 2022) designed a new SSM layer, H3, which further narrowed the performance gap between SSM and Transformers. Recently, (Gu & Dao, 2023) proposed a data-dependent SSM structure and built a universal language model backbone network: Mamba. Its selective mechanism and hardware-aware design allow it to maintain computational efficiency and excellent performance while scaling to billions of parameters.

## A.3    MIXTURE OF EXPERTS

The Mixture of Experts model was first introduced by (Jacobs et al., 1991), which controls a system composed of different networks called experts through a supervisory program, with each expert responsible for handling a specific subset of training samples. (Shazeer et al., 2016) introduced the concept of sparsity into MoE and applied it to LSTM models for translation tasks. With the development of large language models, (Fedus et al., 2022) extensively investigated the stability issues of MoE models during training and fine-tuning processes, and built a MoE model with 16 trillion parameters and 2048 experts. Recently, (Xue et al., 2024) proposed OpenMOE, which further explores the details of MoE using the power of the open-source community, thereby promoting the development of MoE.

## B    OVERALL STRUCTURE OF SINGLE-TASK EEGMAMBA

Figure 7 shows the structure of the single-task EEGMamba model. Compared to EEGMamba, the single-task version removes the MoE modules and the redundant spatial-adaptive convolution branches, retaining only one convolution to process the raw EEG signals. The tokenize layer and BiMamba blocks are kept, with support for stacking any number of BiMamba layers. Additionally, the task-aware classifier in the original EEGMamba is replaced with a standard classifier. Overall, single-task EEGMamba is a lightweight Mamba-based model for EEG classification.

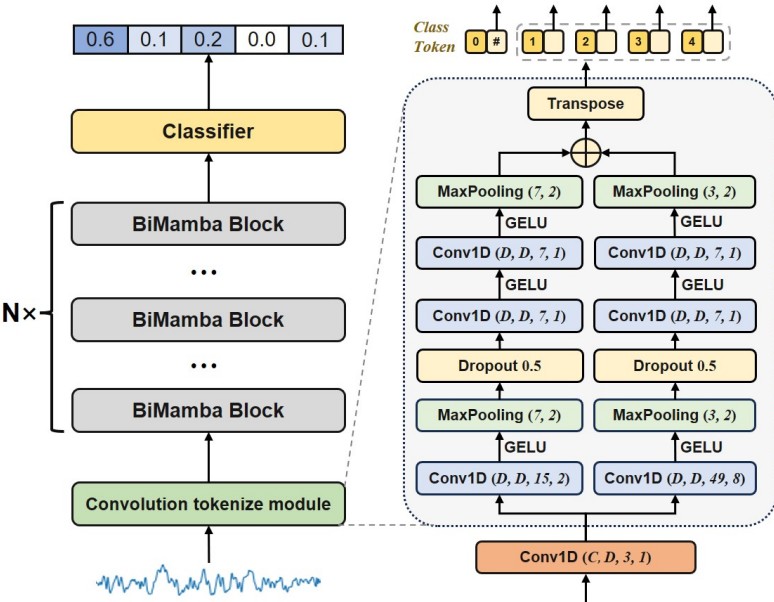

Figure 7: Overall structure of Single-task EEGMamba.

# C   NOTAION TABLE

Table 6 shows the notations used in the main text.

Table 6: Notations used in EEGMamba.

| Symbols | Descriptions |
|---|---|
| $B \in \mathbb{N}^+$ | Batch size |
| $C_i \in \mathbb{N}^+$ | Numbers of channels in EEG signals |
| $D \in \mathbb{N}^+$ | Hidden dimension of the model |
| $L_i \in \mathbb{N}^+$ | Numbers of data points in EEG signals |
| $x \in \mathbb{R}^{B \times C_i \times L_i}$ | EEG signals |
| $CNN_{SA}$ | Spatial-adaptive convolution module |
| $CNN_S$ | Small kernel convolution module |
| $CNN_W$ | Wide kernel convolution module |
| $y_{SA}$ | Features extracted by the spatial-adaptive convolutional module |
| $z_s \in \mathbb{R}^{B \times N_s \times D}$ | Small kernel feature token sequence |
| $z_w \in \mathbb{R}^{B \times N_w \times D}$ | Wide kernel feature token sequence |
| $T \in \mathbb{R}^{B \times (N+1) \times D}$ | EEG token sequence |
| $t_s^j \in \mathbb{R}^{B \times D}$ | Small kernel feature token |
| $t_w^j \in \mathbb{R}^{B \times D}$ | Wide kernel feature token |
| $t_{cls} \in \mathbb{R}^{B \times D}$ | Class token for EEG classification |
| $N_s \in \mathbb{N}^+$ | Numbers of small kernel feature tokens |
| $N_w \in \mathbb{N}^+$ | Numbers of wide kernel feature tokens |
| $N \in \mathbb{N}^+$ | Numbers of overall EEG tokens |
| $Conv_f$ | Forward causal convolution in BiMamba block |
| $Conv_b$ | Backward causal convolution in BiMamba block |
| $SSM_f$ | Forward SSM module in BiMamba block |
| $SSM_b$ | Backward SSM module in Bimamba block |
| $N_e$ | Numbers of experts in MoE |
| $E_i$ | The $i$-th expert in MoE |
| $E^u$ | Universal expert in MoE |
| $G$ | Gating network in MoE |
| $e_i$ | Gating score of the $i$-th expert |
| $\omega$ | Output weight of the universal expert |
| $t_{task} \in \mathbb{R}^{B \times D}$ | Task token for task-aware gating network |
| $L_b$ | Balance loss for loading balance |
| $L_z$ | Router z-loss for training stability |
| $L_{aux}$ | Auxiliary loss for loading balance and training stability |

# D DATASET

## D.1 SIENA SCALP EEG DATABASE

The Siena Scalp EEG Database consists of EEG recordings of 14 patients acquired at the Unit of Neurology and Neurophysiology of the University of Siena. Subjects include 9 males (ages 25-71) and 5 females (ages 20-58). Subjects were monitored with a Video-EEG with a sampling rate of 512 Hz, with electrodes arranged on the basis of the international 10-20 System. Most of the recordings also contain 1 or 2 EKG signals. The data were acquired employing EB Neuro and Natus Quantum LTM amplifiers, and reusable silver/gold cup electrodes. Patients were asked to stay in the bed as much as possible, either asleep or awake. The diagnosis of epilepsy and the classification of seizures according to the criteria of the International League Against Epilepsy were performed by an expert clinician after a careful review of the clinical and electrophysiological data of each patient. In our experiment, we removed non-EEG signals from each EDF record, retaining 29 EEG channels and ensuring that the signals from different subjects maintained the same channel order: Fp1, F3, C3, P3, O1, F7, T3, T5, Fc1, Fc5, Cp1, Cp5, F9, Fz, Cz, Pz, Fp2, F4, C4, P4, O2, F8, T4, T6, Fc2, Fc6, Cp2, Cp6, F10. We discarded the data from Subject 10 due to the lack of some necessary EEG channels. The data records, after channel unification, were segmented into 4-second segments to facilitate classification.

## D.2 CHB-MIT

The CHB-MIT Scalp EEG Database is collected by the Children's Hospital Boston, which contains 24 cases of 23 patients with intractable seizures. The first 23 cases are from 22 patients (17 females, aged 1.5-19 years; 5 males, aged 3-22 years). For the last case, there is no clear gender or age record. the Children's Hospital Boston evaluated the potential conditions for surgical intervention in all epilepsy patients after discontinuing medication for a period of time, and monitored the patients for several days. The original EEG record was obtained using 256 Hz sampling rate with 16-bit resolution from electrodes placed according to the international 10-20 EEG electrode positions and nomenclature (Janjarasjitt, 2017). Given that the number of available channels varies among different patients, we select 23 common channels and discarded data from less than 23 channels. Due to the varying duration of the original data ranging from tens of minutes to several hours, we have truncated it into 4-second segments for easy classification.

## D.3 SLEEPEDF-20

SleepEDF-20 includes Polysomnography (PSG) records from each subject for two consecutive days and nights. The recording of subject 13 on the second night was lost due to a failing cassette or laserdisc. Sleep experts use R&K rules (Wolpert, 1969) to visually determine signal characteristics and label each 30 second period in the dataset as one of eight stages W, N1, N2, N3, N4, REM, MOVEMENT, UNKNOWN. Similar to previous work (Huy et al., 2019), N3 and N4 were merged into N3. In addition, the stages of "MOVEMENT" and "UNKNOWN" have also been removed. (Eldele et al., 2021) have preprocessed the raw data, retaining the Fpz-Cz channel with a sampling rate of 100 Hz, and make it publicly available at `https://researchdata.ntu.edu.sg/dataset.xhtml?persistentId=doi:10.21979/N9/MA1AVG`. We use this version.

## D.4 SHHS

Sleep Heart Health Study (SHHS) is a multi-center cohort study on the cardiovascular and other consequences associated with sleep apnea. The research subjects suffer from various diseases, including lung disease, cardiovascular disease, and coronary heart disease. (Eldele et al., 2021) have preprocessed the raw data, including retaining the C4-A1 channel with a sampling rate of 125 Hz, and make it publicly available at `https://researchdata.ntu.edu.sg/dataset.xhtml?persistentId=doi:10.21979/N9/EAMYFO`. Additionally, in order to reduce the impact of these diseases, only subjects who are considered to have regular sleep patterns (such as subjects with apnea hypopnea index (AHI) less than 5) are retained, and the evaluation criteria here refer to the research method of (Fonseca et al., 2016). Finally, data from 329 participants out of 6441 are retained.

## D.5 DEAP

In the DEAP dataset, movies are used as emotional inducers in experiments. This dataset contains data from over 32 participants aged between 19 and 37, half of whom are females. Participants sit one meter away from the screen. The device records EEG signals at a sampling rate of 512 Hz. 40 selected music video clips were used to trigger emotions. At the end of each video, participants were asked to evaluate their level of arousal, valence, preference, and dominance. The self-assessment scale ranges from 1 to 9. The scores of the subjects are divided into two categories (low or high) based on a stable threshold of 4.5. During the preprocessing process, the EEG signal is downsampled to 128 Hz and a bandpass filter with a cutoff frequency of 4-45 Hz is applied. In this paper, we use the same channel selection as (Khateeb et al., 2021), which includes four electrodes: Fp1, Fp2, F3, and C4.

## D.6 SEED

The SEED dataset collects EEG data from 15 participants while watching emotional movies. It contains a total of 45 experiments. The EEG data is collected by 62 channels based on the international 10-20 system and a sampling rate of 1000 Hz. During the preprocessing process, the data is downsampled to 200 Hz and subjected to a bandpass filter ranging from 0 to 75 Hz. The extraction of EEG sections was based on the duration of each movie, and we further cut these EEG into segments of 20 seconds in length. Within each subject's data file, there are 16 arrays, with 15 of these arrays containing 15 preprocessed segments of EEG data from the experiment. The label array includes corresponding emotional labels, where 1 for positive, 2 for negative, and 3 for neutral emotions.

## D.7 SHU

The motor imagery dataset experiment consists of three phases. The first phase (0-2 seconds) is the resting preparation period, during which subjects can rest, perform minor physical activities, and blink. The second phase (2-4 seconds) is the cue phase, where an animation of left or right hand movement appears on the monitor, indicating the upcoming task. The third phase (4-8 seconds) is the MI (Motor Imagery) phase, during which subjects perform the hand movement MI task as prompted, and EEG signals are recorded. We only use 4 seconds of data from the third phase (i.e. MI stage) for classification. Each session consists of 100 trials, with five sessions conducted for each subject every 2 to 3 days, resulting in a total of 500 trials per subject.

## D.8 BCI-IV-2A

The BCI-IV-2a dataset includes EEG signals obtained from trials involving 9 subjects. This experiment includes four different motor imagery tasks: left hand, right hand, foot, and tongue. Each participant participated in two training sessions, with six sessions per session. In each run, there were 48 trials, a total of 288 trials (12 trials per MI task, a total of 72 trials per task). A set of 25 Ag/AgCl electrodes were used in the experiment, of which 22 were dedicated to recording EEG signals, while the remaining three electrodes recorded eye movement signals (not used in our experiment). All recorded signals are processed through a bandpass filter of 0.5 to 100 Hz and a 50 Hz notch filter. The sampling frequency is set to 250 Hz. Similar to Shu, the experiment consists of three phases, with the EEG from the third phase being used for classification. This EEG data, which is for motor imagery, has a duration of 3 seconds and a sampling frequency of 75 Hz.

## E EXPERIMENTAL RELATED SUPPLEMENTS

### E.1 LOAD BALANCE AND MODEL STABILITY IN MoE

Training an MoE typically encounters two issues: (1) Load imbalance: the gating network tends to select only a few experts. (2) Training instability: excessively large gating values for a few experts lead to an unstable training process. To address these issues, we incorporate balance loss $L_b$ (Shazeer et al., 2016) and router z-loss $L_z$ (Zoph et al., 2022) as auxiliary losses for the model to mitigate load imbalance and training instability, as shown in Equation (11), where $B$ represents the batch size.

$$L_b = \frac{Std(e(T))}{Mean(e(T))} \tag{11}$$

$$L_z = \frac{1}{B} \sum_{i=1}^{B} (log(exp(T)))^2$$

$$L_{aux} = L_b + L_z$$

### E.2 TASK-AWARE CLASSIFIER

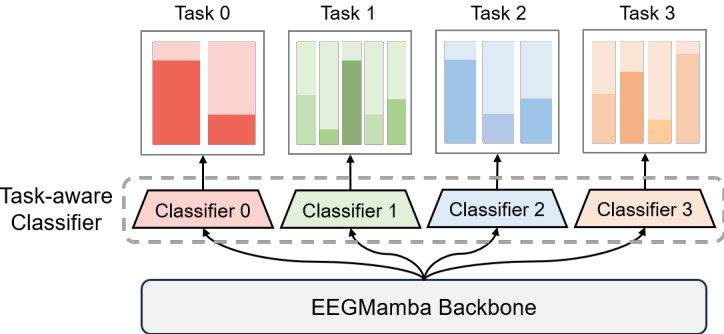

Figure 8: Overall structure of Task-aware Classifier.

To address the inconsistency in the number of classes, we introduce a task-aware classifier, consisting of sub-modules, each with a single linear layer configured to have a different number of output dimension corresponding to the specific number of classes, as shown in Figure 8. This approach enables uniform processing of EEG data with varying class counts. The number of classes for each dataset is pre-defined, and for data belonging to the same task, the task identifier is passed through the forward pass, ensuring that data from the same task produce outputs with consistent shapes.

Let $t_{cls} \in \mathbb{R}^{B \times D}$ represents the class token output from the final task-aware MoE block. As shown in Equation 12, $logits_i$ is the result obtained through task-aware classifier, where the output dimension is changed from the number of classes $K_i$ determined by the task $i$.

$$logits_i = Linear_i(t_{cls}) \in \mathbb{R}^{B \times K_i} \tag{12}$$

### E.3 SUBJECT DIVISION IN EEGMAMBA EXPERIMENT

Table 7 presents the grouping and combination of subjects in our five-fold cross-validation experiment. The numbers in the table represent subject IDs in the dataset. Generally, '1 ∼ 5' indicates five subjects, including subject 1 through subject 5. For the SHHS dataset, only a subset of subjects is used (D.4), and '10 - 2021' refers to all selected subjects within the range of IDs from 10 to 2021, rather than all subjects in that range consecutively.

Table 7: Division and combination of subjects in different datasets.

| Group | Epilepsy detection | | Sleep stage classification | | Emotion recognition | | Motor imagery | |
|---|---|---|---|---|---|---|---|---|
| | Siena | CHB-MIT | SleepEDF-20 | SHHS | DEAP | SEED | Shu | BCI-IV-2a |
| 1 | 0, 1, 3 | $1 \sim 5$ | $0 \sim 3$ | 10 - 1021 | $1 \sim 6$ | $1 \sim 3$ | $1 \sim 5$ | 1, 2 |
| 2 | 5, 6, 7 | $6 \sim 10$ | $4 \sim 7$ | 1023 - 2956 | $7 \sim 12$ | $4 \sim 6$ | $6 \sim 10$ | 3, 4 |
| 3 | 9, 11, 12 | $11 \sim 15$ | $8 \sim 11$ | 2983 - 4047 | $13 \sim 18$ | $7 \sim 9$ | $11 \sim 15$ | 5, 6 |
| 4 | 13, 14, 15 | $16 \sim 19$ | $12 \sim 15$ | 4051 - 4781 | $19 \sim 25$ | $10 \sim 12$ | $16 \sim 20$ | 7, 8 |
| 5 | 16, 17 | $20 \sim 23$ | $16 \sim 19$ | 4783 - 5789 | $26 \sim 32$ | $13 \sim 15$ | $21 \sim 25$ | 9 |
| Total | 13 | 23 | 20 | 329 | 32 | 15 | 25 | 9 |

### E.4 TRAINING STRATEGY

Training the EEGMamba model across multiple EEG datasets with varying tasks presents two primary challenges. First, the inconsistency in the number of channels and lengths across different EEG datasets prevents direct mixed-batch training. Second, training different datasets sequentially may lead to the model forgetting knowledge from earlier datasets.

To address these issues, we propose a dynamic sampling training strategy. Specifically, in each training iteration, we randomly select a batch from the same dataset based on the proportion of samples that have not yet participated in the training. This ensures that data within the same batch have consistent channel counts and lengths. Furthermore, as the probability of sampling each dataset is dynamically adjusted based on the amount of untrained data, larger datasets receive more attention at the beginning of training, while smaller datasets are primarily sampled later, effectively avoiding the model's forgetting of smaller datasets.

### E.5 PARAMETER SETTINGS

Table 8 shows the important hyperparameters we used in the experiment.

Table 8: Hyperparameters for EEGMamba.

| Hyperparameters | EEGMamba | Single-task EEGMamba |
|---|---|---|
| Hidden dimension | 128 | 128 |
| BiMamba layers | 8 | 2 |
| MoE blocks | 8 | None |
| Experts | 8 | None |
| Experts activated each time | 2 | None |
| Batch size | 128 | 128 |
| Learning rate | 2e-4 | 2e-4 |
| Optimizer | Adamw | Adamw |
| Weight decay | 1e-6 | 1e-6 |
| Training epochs | 100 | 100 |

### E.6 METRICS

**Accuracy** is a fundamental performance metric for classification models, defined as the ratio of correctly classified samples to the total number of samples. It applies to both binary and multi-class tasks.

**AUROC** is a key metric for evaluating the performance of classification models, summarizing the model's ability to distinguish between positive and negative classes across various thresholds by calculating the area under the ROC curve. The AUROC value ranges from 0 to 1, with a value closer to 1 indicating better classification performance.

**F1 Score** is the harmonic mean of precision and recall, particularly useful in scenarios where a balance between these two metrics is desired. Weighted F1 is used for both binary and multi-class classification in this paper, representing a weighted average of the individual F1 scores for each class, where each score is weighted according to the number of samples in that specific class.

# F BASELINES

We consider the following representative models:

(i) **EEGNet** (Lawhern et al., 2018) is a classic EEG classification network based on depthwise separable convolution, which has a concise structure and demonstrated stable and robust performance in various EEG classification tasks.

(ii) **AttnSleep** (Eldele et al., 2021) is a deep learning model based on the attention mechanism, designed to automatically classify sleep stages by processing polysomnography (PSG) data including EEG.

(iii) **EEG Conformer** (Song et al., 2022) utilizes convolution modules and self-attention modules to capture local features and global dependencies in EEG signals respectively, enabling precise analysis of EEG data.

(iv) **BIOT** (Yang et al., 2023) is a pre-trained model that can be applied to various biosignals include EEG.

(v) **HCANN** (Ji et al., 2024) is a recently proposed EEG classification network featuring a multi-head mechanism that is adaptively modified for EEG signals. It has achieved state-of-the-art (SOTA) performance across three BCI tasks.

We conduct all baseline tests using publicly available pretrained weights and the open-source code. Generally, we use the same training hyperparameters as in Table 7 in the baseline experiments.

## G  VISUALIZATION OF FEATURES EXTRACTED BY SINGLE-TASK EEGMAMBA

Figure 9 shows t-distributed stochastic neighbor embedding (t-SNE) plots of features extracted by single-task EEGMamba from different datasets. The plot exhibits distinct distances between features of different classes and small distances within the same class, indicating the successful extraction of features from different classes by single-task EEGMamba. This may indicate its comprehensive performance superiority across different datasets.

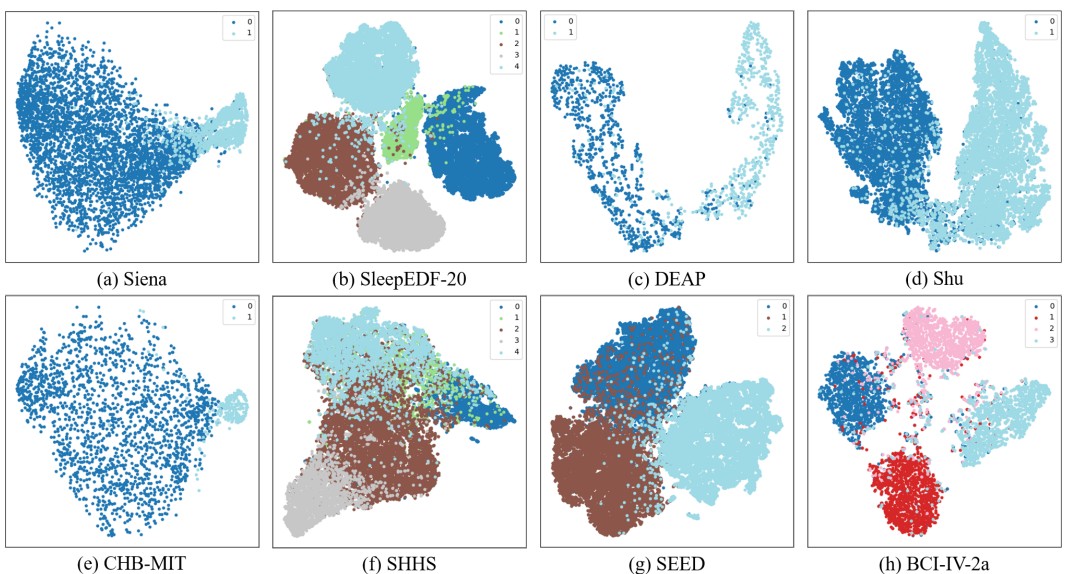

| (a) Siena | (b) SleepEDF-20 | (c) DEAP | (d) Shu |
| (e) CHB-MIT | (f) SHHS | (g) SEED | (h) BCI-IV-2a |

Figure 9: Visualization results of feature extracted by single-task EEGMamba on different datasets.

## H  VISUALIZATION OF MoE WEIGHTS

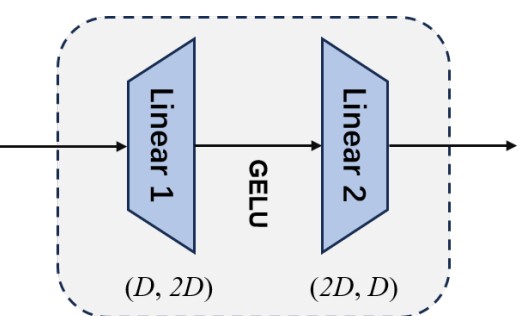

Figure 10: The specific structure of an expert.

In Figure 2, each expert is essentially a Multi-Layer Perceptron (MLP) consisting of two linear layers. The detailed structure is shown in Figure 10, where hidden dimension $D = 128$. We visualize the expert weight heatmap for the final MoE module of EEGMamba, where Figure 11 shows the weights of the first linear layer and Figure 12 shows those of the second linear layer. Clearly, the weight distributions vary across different experts, demonstrating that they specialize in handling different tasks.

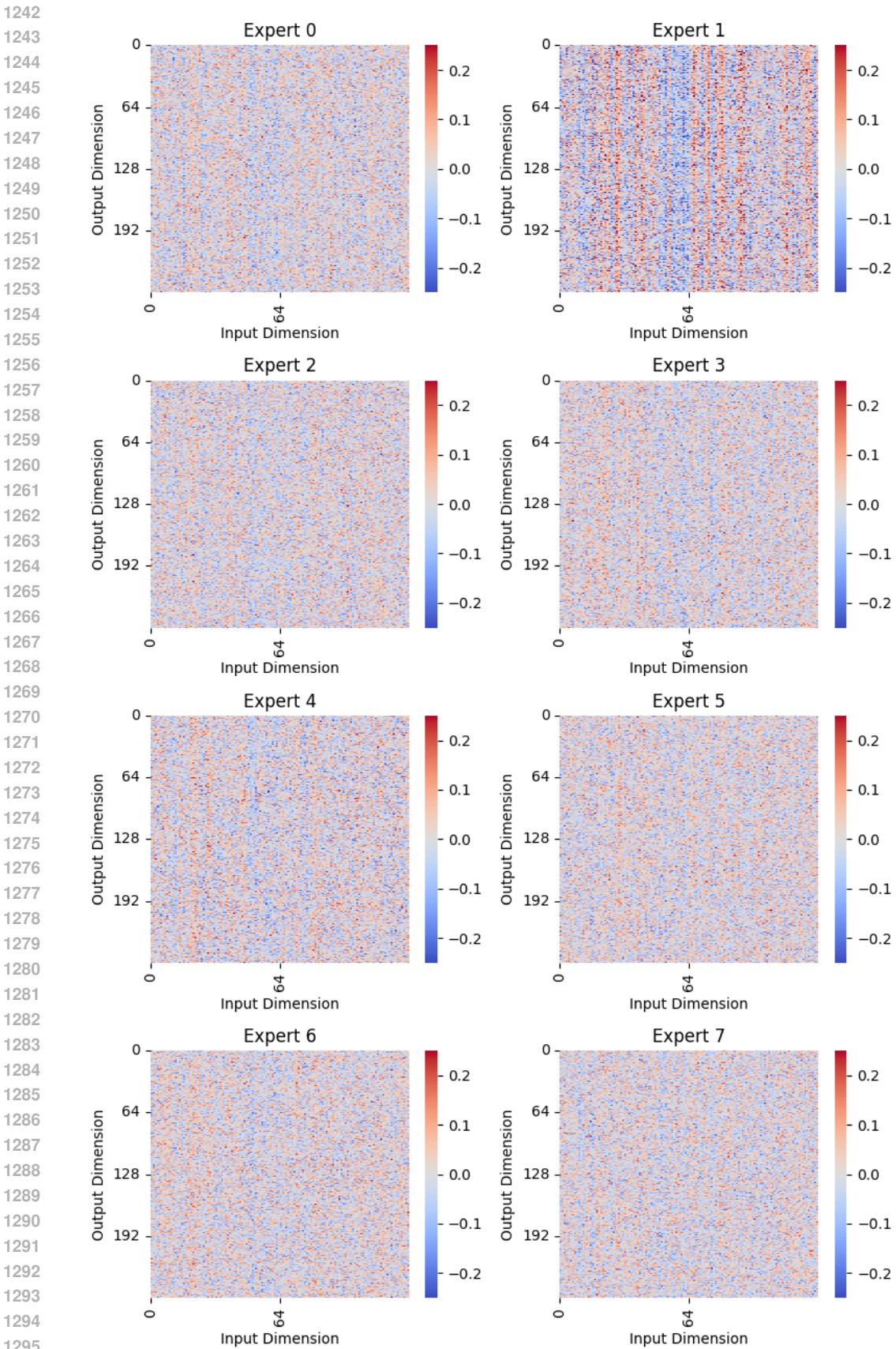

Figure 11: The first linear layer weight visualization of experts in final MoE module.

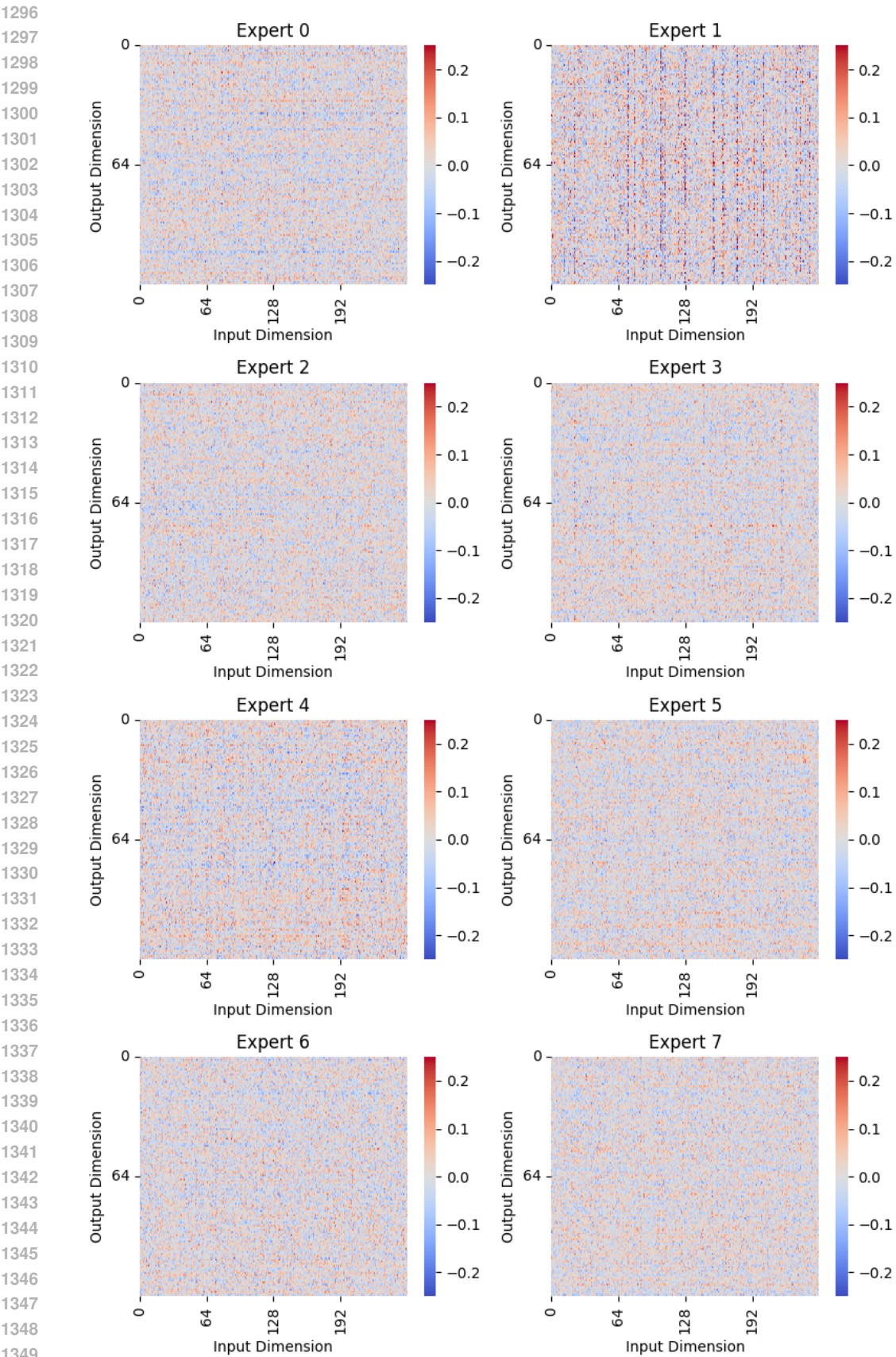

Figure 12: The second linear layer weight visualization of experts in final MoE module.

# I    DETAILED RESULTS ON MEMORY-USAGE AND INFERENCE SPEED

Table 9, 10 present the detailed results of memory-usage and inference speed, where OOM indicates out of memory.

Table 9: Detailed results on Memory-Usage and Inference Speed with single-channel data.

| Sequence Length | | 2000 | 3000 | 5000 | 10000 | 20000 | 40000 |
|---|---|---|---|---|---|---|---|
| EEGNet | Memory-Usage | 646 MiB | 724 MiB | 834 MiB | 1096 MiB | 1608 MiB | 2648 MiB |
| | Inference Speed | 427.20 iter/s | 289.08 iter/s | 174.92 iter/s | 88.11 iter/s | 44.08 iter/s | 21.95 iter/s |
| AttnSleep | Memory-Usage | 3518 MiB | 6670 MiB | 16534 MiB | 61012 MiB | OOM | OOM |
| | Inference Speed | 133.97 iter/s | 81.98 iter/s | 42.65 iter/s | 10.87 iter/s | OOM | OOM |
| EEG Conformer | Memory-Usage | 3748 MiB | 6958 MiB | 16384 MiB | 62702 MiB | OOM | OOM |
| | Inference Speed | 104.92 iter/s | 56.14 iter/s | 22.28 iter/s | 5.89 iter/s | OOM | OOM |
| HCANN | Memory-Usage | 2340 MiB | 3318 MiB | 3936 MiB | 9868 MiB | 21108 MiB | 49724 MiB |
| | Inference Speed | 92.18 iter/s | 62.90 iter/s | 37.84 iter/s | 17.83 iter/s | 8.21 iter/s | 3.87 iter/s |
| Single-task EEGMamba | Memory-Usage | 2864 MiB | 3936 MiB | 6202 MiB | 11600 MiB | 22938 MiB | 45174 MiB |
| | Inference Speed | 101.43 iter/s | 81.78 iter/s | 46.49 iter/s | 21.21 iter/s | 10.15 iter/s | 5.30 iter/s |

Table 10: Detailed results on Memory-Usage and Inference Speed with multi-channel data.

| Sequence Length | | 2000 | 3000 | 5000 | 10000 | 20000 | 40000 |
|---|---|---|---|---|---|---|---|
| EEGNet | Memory-Usage | 1630 MiB | 2014 MiB | 2804 MiB | 4810 MiB | 8938 MiB | 17026 MiB |
| | Inference Speed | 285.19 iter/s | 191.88 iter/s | 115.93 iter/s | 58.21 iter/s | 29.22 iter/s | 14.65 iter/s |
| AttnSleep | Memory-Usage | 3532 MiB | 6682 MiB | 16554 MiB | 61028 MiB | OOM | OOM |
| | Inference Speed | 140.96 iter/s | 79.58 iter/s | 41.66 iter/s | 10.81 iter/s | OOM | OOM |
| EEG Conformer | Memory-Usage | 6838 MiB | 11590 MiB | 24430 MiB | 63650 MiB | OOM | OOM |
| | Inference Speed | 63.44 iter/s | 36.58 iter/s | 16.58 iter/s | 4.65 iter/s | OOM | OOM |
| HCANN | Memory-Usage | 27378 MiB | 68034 MiB | OOM | OOM | OOM | OOM |
| | Inference Speed | 44.07 iter/s | 5.18 iter/s | OOM | OOM | OOM | OOM |
| Single-task EEGMamba | Memory-Usage | 2954 MiB | 4140 MiB | 6410 MiB | 12208 MiB | 23758 MiB | 46800 MiB |
| | Inference Speed | 97.31 iter/s | 75.74 iter/s | 43.16 iter/s | 19.72 iter/s | 9.49 iter/s | 5.05 iter/s |

## J  LIMITATIONS

Although the current experimental results show that EEGMamba can be well applied to EEG multi-task classification, it still has some limitations. On the one hand, this paper only covers four kinds of EEG tasks to verify the performance of EEGMamba, which is only a small part of the tasks that EEG can accomplish. On the other hand, it should be extended to other one-dimensional time signals besides EEG to prove the universality of the model in one-dimensional time signals.

