# OpenReview forum: "EEGMamba: Bidirectional State Space Model with Mixture of Experts for EEG Multi-task Classification"
_ICLR.cc/2025/Conference — Submitted to ICLR 2025_

### Official Review · Reviewer_SRQL · 2024-10-21

**Soundness:** 2
**Presentation:** 3
**Contribution:** 2
**Rating:** 3
**Confidence:** 4

**Summary:**

To address the issues of quadratic computational complexity in handling long-term dependencies in EEG classification models, and the lack of cross-task generalization as most models are designed for single tasks, this paper proposes the EEGMamba model. The model introduces the ST-Adaptive module to address the problem of varying EEG signal lengths and channel numbers across different datasets. It also proposes the Bidirectional Mamba to improve computational efficiency and incorporates the MoE (Mixture of Experts) module to simultaneously capture both the commonalities and differences in EEG signals. Relevant experiments were conducted on eight datasets across four EEG tasks.

**Strengths:**

This paper presents a novel multi-task EEG classification model that incorporates the Bidirectional Mamba and MoE modules, offering a structurally innovative approach. Experiments were conducted on eight datasets, covering a wide range of downstream EEG tasks. The paper is clearly written and easy to follow, with a particularly well-explained method section.

**Weaknesses:**

This paper proposes a multi-task EEG classification model aimed at performing classification across multiple datasets (downstream tasks) using a single model. However, the experimental results do not demonstrate a clear advantage over other single-task models, making it difficult to convincingly argue for the benefits and necessity of the multi-task approach. Additionally, the ablation study results for the various modules lack significant and consistent differences, making it challenging to prove the effectiveness of each module. Moreover, the motivation for using the Bidirectional Mamba is insufficiently justified. The ST-Adaptive method, proposed to address the issue of varying EEG channel numbers across datasets, is essentially a module integration, lacking in innovation.
1. In this work, the ST-Adaptive module applies a different one-dimensional convolution for each task, transforming the varying original channel numbers into a fixed number of $D$ channels. However, this approach does not seem to fully achieve the concept of being "adaptive." If a new task emerges with a number of channels outside the predefined range of $C_0$ to $C_N$ in the model, how would this be addressed? This raises concerns about the generalizability and flexibility of the current method in handling unforeseen tasks with different channel configurations.
2. The purpose of using Mamba in this paper is to reduce computational complexity, conserve resources, and improve efficiency. Generally, the ultimate goal of improving efficiency in EEG models is to achieve real-time recognition. Given that EEG signals, like natural language, possess temporal characteristics, theoretically, a unidirectional Mamba would better meet this requirement, as it only requires past data rather than future information. The motivation for employing Mamba, especially Bidirectional Mamba, in this work is not sufficiently clear or logically aligned with this objective.
3. In Section 4.1, the authors present the performance of the single-task EEGMamba and other transformer-based models concerning memory usage and inference speed as the sequence length increases. However, it is not clearly evident from Figure 4 that EEGMamba demonstrates a significant advantage over the other methods. Additionally, while it is acknowledged that memory usage increases and inference speed decreases with longer sequence lengths for both single-channel and multi-channel scenarios, the authors do not specify the actual sequence lengths employed in the current eight EEG tasks. This omission lacks a reference point, making it difficult to ascertain whether EEGMamba exhibits superior performance. Furthermore, as indicated in Appendix I, EEGNet appears to perform better in terms of memory usage and inference speed, while also demonstrating commendable performance across various datasets. This further undermines the effectiveness of the proposed method in this paper.
4. In Section 4.2, the authors present the performance of EEGMamba in multi-task classification. However, I observe that EEGMamba does not demonstrate a significant advantage over the baseline models, and in many datasets, its performance is inferior to that of other single-task models, indicating that the multi-task approach does not facilitate mutual enhancement among tasks. Therefore, I question the necessity of employing a single model to address multiple tasks rather than utilizing several smaller models for different tasks, which might yield better results. The existing findings lack persuasiveness and do not adequately support the motivations for this work or the claims regarding the strong generalization capabilities of the proposed multi-task model.
5. Regarding Figure 5, I observe that, apart from the sleep stage task, where there is a considerable variation in the activation probabilities of different experts, the activation probabilities for the other tasks across the eight experts are generally quite uniform. This uniformity makes it challenging to demonstrate a particular preference for any specific expert. How can the effectiveness and necessity of the MoE approach be substantiated under these circumstances?
6. Figure 6 presents the ablation study results for the various modules of EEGMamba. However, the data indicate that these modules appear to have minimal discernible impact, as the experimental results across the Siena, CHB-MIT, and SHHS datasets show little variation. This raises concerns regarding the ability to substantiate the effectiveness of each module.

**Questions:**

1. Could you please specify how the EEG tasks are encoded into task tokens?
2. I noticed that the DEAP dataset was utilized in this study, but only data from four electrodes were selected. What is the rationale behind this choice? Additionally, regarding the binary classification on the DEAP dataset, does it pertain to valence, arousal, liking, or dominance? Furthermore, in Table 1, the authors provide the optimal segment lengths for all datasets. What references were used to determine these durations? I observed that, unlike most existing works that employ shorter segments of 1s, 2s, or 4s for the DEAP and SEED datasets, this paper utilizes segment lengths of 60s and 20s. What is the reasoning for selecting such unusually long data lengths?
3. This work employs five-fold cross-validation for data partitioning, which does not appear to be a commonly used EEG dataset partitioning method. What is the rationale or basis for this choice?

---

> ### Author Response · Authors · 2024-11-23
> **Reply to W1-W2**
>
> Thank you for your detailed and thoughtful review of our manuscript. We value your comments and suggestions, and we are grateful for the opportunity to address them. In this response, we will provide further clarification and empirical evidence to address the concerns you raised. Here is our detailed response:
>
> **W1: The generalizability and flexibility of the current method in handling unforeseen tasks with different channel configurations.**
> Our model can be extended to new datasets after training. We can describe the specific extension from two different situations as follows:
> - When the newly added dataset has the same number of tasks, channels and classes as the previous dataset, we can directly use the task index of the original dataset. For example, the two datasets SleepEDF20 and SHHS used in the manuscript experiment can be replaced with each other, that is, if you want to apply the model trained on SleepEDF20 to SHHS, you only need to encode the same task and make a few epochs of fine-tuning.
> - When the newly added dataset cannot meet the conditions in 1, we need to give it a new task number and pre-set its number of channels and classes, and then a few epochs of fine-tuning. This is broadly similar to what most current foundation models do, except that we need to pre-set the task number and the number of channels.
>
> We use the Confused student EEG brainwave data [1] (hereinafter referred to as Confused EEG), which is a completely new task for EEGMamba. We applied the existing weights (i.e., the weights corresponding to Table 2-5 in the manuscript) to the Confused EEG, using 4 different random numbers to obtain a 7:3 training-test set ratio, similar to the approach in [3] (while [2] used five-fold cross-validation). We trained for 10 epochs each time and averaged the results. The test results are shown as follows:
>
> | Classification   Network | [2], 2019 | [3], 2023 | EEGMamba |
> |:------------------------:|:---------:|:---------:|:--------:|
> |    Reported   Accuracy   |   0.7500  |   0.7670  |    0.7825   |
>
> [1] Wang H, Li Y, Hu X, et al. Using EEG to Improve Massive Open Online Courses Feedback Interaction[C]//AIED workshops. 2013.
> [2] Wang H, Wu Z, Xing E P. Removing confounding factors associated weights in deep neural networks improves the prediction accuracy for healthcare applications[C]//Pacific Symposium on Biocomputing. Pacific Symposium on Biocomputing. NIH Public Access, 2019, 24: 54.
> [3] Lim Z Y, Neo Y L. Confused vs Non-Confused Electroencephalography Signal Classification Using Deep Learning Algorithm[C]//2023 IEEE International Conference on Automatic Control and Intelligent Systems (I2CACIS). IEEE, 2023: 195-200.
>
> **W2: The motivation for employing Mamba, especially Bidirectional Mamba, in this work is not sufficiently clear or logically aligned with this objective.**
> In this work, we use the bidirectional Mamba for feature extraction from EEG signals, not only to reduce computational complexity but also to enhance the model’s overall ability to capture the characteristics of EEG signals.
> **Let us be crystal clear: adopting bidirectional modeling does not affect the real-time detection capabilities of EEG signals.** In reality, whether it’s single-directional or bidirectional modeling, what we’re processing are already sampled EEG signal segments—not like in natural language tasks, where you need to predict the next word. Therefore, bidirectional modeling has no negative impact on real-time performance.
> As we know, Transformer-based models typically suffer from quadratic computational complexity, which leads to significant performance bottlenecks in practical applications. Take medical diagnostic scenarios, for example, such as sleep disorder detection or Alzheimer’s diagnosis, where you need to process long durations (e.g., several minutes) and a large number of channels (e.g., 62 channels) of EEG data. In such cases, Transformer-based models consume an enormous amount of GPU memory, making it nearly impossible to meet the real-time processing requirements. On the other hand, the Mamba model, with its linear computational complexity, consumes much less memory when handling long sequences, making it far better suited for high-load tasks like these. This is the primary reason we chose the Mamba model.
> The reason for choosing bidirectional Mamba is that the original Mamba model was designed for language generation tasks, which typically use single-directional modeling. But the nature of EEG signals makes single-directional modeling unsuitable—it could lead to the loss of earlier information during the scanning process. To more comprehensively and accurately capture the temporal features of EEG signals, we employed a bidirectional Mamba structure. Our ablation experiments also well demonstrated the performance difference between single-directional modeling and bidirectional modeling in EEG classification.

---

> ### Author Response · Authors · 2024-11-23
> **Reply to W3-W6**
>
> **W3: In Section 4.1, the authors present the performance of the single-task EEGMamba and other transformer-based models concerning memory usage and inference speed as the sequence length increases…**
> From Figure 4, it is clear that, for other Transformer-based models, the blue line representing the Single-task EEGMamba shows a distinct advantage. The only model that can somewhat match Single-task EEGMamba when processing single-channel data is HCANN, but in multi-channel scenarios, its performance rapidly declines. Meanwhile, all the other models consistently perform worse than Single-task EEGMamba.
> As for the signal length, the actual sequence length for each signal can be obtained from the rate × duration in Table 1. We will consider replacing "duration" with "sequence length" to make it easier for the readers to understand.
> Clearly, EEGNet’s overall performance on the dataset we’re using is far from comparable to EEGMamba, especially when it comes to tasks like sleep stage detection, which involves longer signals. Additionally, EEGNet requires dedicated training for each dataset, which is quite a tedious process.
>
> **W4: In Section 4.2, the authors present the performance of EEGMamba in multi-task classification…**
> EEGMamba has been evaluated on eight publicly available EEG datasets across four different tasks, demonstrating superior performance in seizure detection, emotion recognition, sleep stage classification, and motor imagery. Our model ranks among the top three on seven datasets and achieves the best performance on four datasets, outperforming existing state-of-the-art (SOTA) models across multiple datasets.
> In addition, EEGMamba is an end-to-end system that does not require separate pre-training and fine-tuning stages, offering stronger generalization ability than pre-trained models. Unlike other classification networks that require multiple training sessions with manual adjustments to data length, channel count, and class numbers, EEGMamba only needs to be trained once to achieve these results.
>
> **W5: Regarding Figure 5, I observe that, apart from the sleep stage task…**
> When it comes to the activation probabilities of experts, it’s not only important to observe the differences in activation probabilities between experts for a given task, but even more crucial to focus on the tendency of experts being activated across different tasks. For instance, in the task of seizure detection, experts 5 and 6 appear to be more easily activated, while for tasks like emotion recognition and motor imagery, experts 2 and 4 are more likely to be chosen, respectively. This suggests that different tasks tend to favor different experts. In other words, each task seems to have its own preference for specific experts, which is an interesting characteristic that could be leveraged to improve task-specific performance.
>
> **W6: Figure 6 presents the ablation study results for the various modules of EEGMamba…**
> We sincerely apologize, but due to the particularly poor performance of certain variants (such as the unidirectional Mamba), it’s been difficult to further adjust the y-axis range in a way that would make the differences more visually distinct. Additionally, for some datasets and tasks, the inherent characteristics of the dataset (for example, the extreme class imbalance in the epilepsy dataset) make it quite challenging to observe significant differences when evaluating using accuracy. We recommend that you enlarge the image up to 400% so that you’ll be able to more clearly see the differences between the various variants.

---

> ### Author Response · Authors · 2024-11-23
> **Reply to Q1-Q3**
>
> **Q1: Could you please specify how the EEG tasks are encoded into task tokens?**
> **A1:** We sincerely apologize for any confusion caused by the unclear presentation. The task token is pre-assigned to each task before model training begins. When the EEG feature tokens pass through the task-aware gate, both the task token and the EEG feature tokens are both involved in the computation of the task-aware gate.
>
> **Q2: I noticed that the DEAP dataset was utilized in this study, but only data from four electrodes were selected… What is the reasoning for selecting such unusually long data lengths?**
> **A2:** As described in Appendix D5, we selected four channels for classification based on the approach outlined in [1], where grid search was used to achieve similar results. More detailed information on this procedure can be found directly in [1].
> Thank you for pointing out that our manuscript omitted a description of the emotional evaluation dimension used. Specifically, we employed valence as the indicator for binary classification, and we will include this information in the revised version of the paper.
> Regarding the selection of duration, we found that previous literature does not provide direct evidence supporting the choice of 1s, 2s, or 4s as the optimal segment lengths [2][3]. It is likely that these choices were made due to limitations in Transformer-based models, which struggled to classify longer EEG sequences due to constrained computational resources.
>
> [1] Khateeb M, Anwar S M, Alnowami M. Multi-domain feature fusion for emotion classification using DEAP dataset[J]. IEEE Access, 2021, 9: 12134-12142.
> [2] Song Y, Zheng Q, Liu B, et al. EEG conformer: Convolutional transformer for EEG decoding and visualization[J]. IEEE Transactions on Neural Systems and Rehabilitation Engineering, 2022, 31: 710-719.
> [3] Jiménez-Guarneros M, Fuentes-Pineda G. Cross-subject EEG-based emotion recognition via semi-supervised multi-source joint distribution adaptation[J]. IEEE Transactions on Instrumentation and Measurement, 2023.
>
> **Q3: This work employs five-fold cross-validation for data partitioning, which does not appear to be a commonly used EEG dataset partitioning method. What is the rationale or basis for this choice?**
> **A3:** We apologize for any confusion, but as far as we know, subject-wise cross-validation is a widely used method for partitioning EEG datasets, with common approaches including five-fold, ten-fold, or even twenty-fold cross-validation. References supporting this include [1][2][3][4][5]. We are curious to know the rationale behind your statement that this is an uncommon form of data partitioning.
>
> [1] Eldele, E., Chen, Z., Liu, C., Wu, M., Kwoh, C. K., Li, X., & Guan, C. (2021). An attention-based deep learning approach for sleep stage classification with single-channel EEG. IEEE Transactions on Neural Systems and Rehabilitation Engineering, 29, 809-818.
> [2] Yuan Z, Zhang D, Chen J, et al. Brant-2: Foundation Model for Brain Signals[J]. arXiv preprint arXiv:2402.10251, 2024.
> [3] Ji Y, Li F, Fu B, et al. A novel hybrid decoding neural network for EEG signal representation[J]. Pattern Recognition, 2024, 155: 110726.
> [4] Lawhern V J, Solon A J, Waytowich N R, et al. EEGNet: a compact convolutional neural network for EEG-based brain–computer interfaces[J]. Journal of neural engineering, 2018, 15(5): 056013.
> [5] Zhang Z, Zhong S, Liu Y. TorchEEGEMO: A deep learning toolbox towards EEG-based emotion recognition[J]. Expert Systems with Applications, 2024, 249: 123550.

---

> > ### Comment · Reviewer_uVYA · 2024-11-25
> >
> > Thank you for the clarification. Based on the additional information provided by the authors, I have decided to raise the score to 6. The paper introduces some novel and intriguing model design ideas, such as the use of task tokens to specify downstream tasks and the simultaneous learning with multiple datasets. However, the following concerns remain:
> >
> > 1. Gap Between Offline Analysis and Real-Life Application
> > While the use of task tokens during offline analysis is conceptually straightforward—since the researcher knows which task (or dataset) is being evaluated—the applicability in real-life scenarios remains unclear. For instance, in a hypothetical situation involving simultaneous seizure and emotion monitoring alongside the control of a motor-imagery-based BCI (although admittedly unrealistic, this scenario reflects the diversity of the benchmarked datasets in this study), it is uncertain how task tokens would be assigned in practice. In real-world applications, where the timing and nature of a task are not explicitly known, task token assignment could pose a significant challenge. While it is theoretically possible to assign multiple task tokens simultaneously and generate corresponding outputs, the potential advantages of this approach over traditional methods have not been adequately demonstrated.
> >
> > 2. Relevance of Long Sequence Modeling in EEG
> > The utility of long sequence modeling in EEG analysis appears to be task-specific. For applications such as sleep stage classification and epilepsy detection, long sequence modeling is relevant. However, for other tasks, such as motor imagery, the proposed model may not necessarily outperform existing approaches, such as the EEG Conformer. The benefits of multi-task learning for such cases remain less evident.
> >
> > 3. Lack of Model Interpretability
> > The paper does not sufficiently address the interpretability of the proposed model. It would be highly valuable to analyze what has been learned across datasets in the context of multi-task learning and how this differs from traditional single-dataset approaches. Such an analysis could provide deeper insights into the generalization capabilities of the model and reveal the specific types of EEG features it captures. These insights could significantly enhance the understanding of the model's contributions to EEG analysis.

---

> > > ### Author Response · Authors · 2024-11-25
> > > **Responses and thanks**
> > >
> > > We would like to sincerely thank the reviewer uVYA for insightful comments and for raising important points regarding the applicability of our proposed model. We appreciate the time and effort taken to evaluate our work, and we have carefully considered the feedback provided:
> > > 1. We agree that while task tokens are straightforward in offline analysis, real-world application is more complex. Assigning task tokens in scenarios where task timing is not explicit could indeed be challenging. We sincerely thank the reviewers for providing such insights to help us further improve the manuscript. In future work, we plan to explore dynamic assignment of task tokens during real-time inference and investigate strategies for handling such complexities in practical settings, especially in multi-task environments.
> > > 2. We acknowledge that long sequence modeling is more relevant for certain tasks (e.g., sleep stage classification, epilepsy detection). For tasks like motor imagery, the benefits of long sequence modeling are less clear. Moving forward, we will conduct additional experiments comparing our approach with models like EEG Conformer for different EEG tasks to better understand where our method provides clear advantages.
> > > 3. We appreciate the reviewer’s concern about model interpretability. To address this, we plan to include feature visualizations in future revisions to provide deeper insights into the learned representations and how multi-task learning influences generalization across different datasets. This will help improve the interpretability of the model and its application in real-world settings.
> > >
> > > Once again, we would like to thank the reviewer for their thoughtful and constructive comments. We believe that incorporating these suggestions will significantly strengthen the paper and provide a clearer understanding of the potential and limitations of our approach.

---

> ### Author Response · Authors · 2024-12-02
> **A reminder to reviewerSRQL**
>
> Dear Reviewer SRQL,
>
> We would like to gently remind you that we have submitted our response to the concerns you raised. We hope these further explanations help to resolve any outstanding issues. We sincerely hope that you might consider revising your score based on our response.  If you have any questions, we would be more than happy to discuss with you.
>
> Thank you again for the time and effort you put into reviewing our manuscript.
>
> Best regards,
>
> The authors

---

### Official Review · Reviewer_uVYA · 2024-10-25

**Soundness:** 3
**Presentation:** 3
**Contribution:** 2
**Rating:** 5
**Confidence:** 4

**Summary:**

The authors introduce EEGMamba, a model designed for multi-task EEG classification. EEGMamba consists of an ST-Adaptive module that learns spatial filters for each task, transforming EEG inputs with varying channel counts into a uniform feature space. The module then tokenizes the data using both small and large kernels to capture short-term and long-term features. These tokens are processed by a BiMamba backbone with task-aware Mixture of Experts (MoE) layers, enabling the model to capture both task-specific and shared features. Finally, each task has a dedicated classification head.

EEGMamba allows for multi-task EEG classification in a single training session. The authors evaluated the model’s performance against five other models across eight public datasets, covering tasks such as epilepsy detection, sleep stage classification, emotion recognition, and motor imagery. The experiments used a 5-fold cross-validation approach, where specific subjects were reserved for the test set in each fold. Results show that EEGMamba outperformed competing models under this evaluation, and it demonstrated efficient memory usage and inference speed, particularly with long-sequence data.

**Strengths:**

1. The authors successfully introduce the new Mamba architecture to EEG decoding, achieving strong results across 8 public datasets.

2. Mamba demonstrates memory efficiency and fast inference, making it advantageous for real-world applications.

3. The model effectively addresses the multi-task classification problem, showcasing the feasibility of training a single model for multiple downstream tasks.

**Weaknesses:**

1. The motivation for multi-task training is somewhat unclear. The authors should clarify why multi-task training is necessary and how it can be beneficial for specific applications.

2. Certain questions remain unanswered regarding the practical use of the proposed model, which limits its potential impact. For instance, under the current evaluation scheme, it is unclear how the model would generalize to new datasets or subjects, particularly when new datasets vary in channel count. Does introducing a new dataset require retraining or additional multi-

task training even if it is single-task? Additionally, what would be the training strategy for developing a subject-dependent model within the multi-task framework if only one task is available from the subject? To address this, the authors could consider testing the model on an additional dataset to evaluate whether the pre-trained model can transfer effectively. If it cannot, are there still advantages to the EEGMamba multi-task approach?

3. The authors should discuss the relatively low classification accuracy on the motor imagery datasets, which is currently too low for practical motor imagery classification. If this is due to the evaluation setting, additional experiments with per-subject models should be conducted to assess performance, and these results should be compared to other models.

4. Statistical tests are needed to confirm whether the observed differences between models or modules are significant. For instance, statistical analysis should be conducted for the results in Figure 1 comparing EEGMamba and Single-task EEGMamba, as well as for the different ablation models in Figure 6.

**Questions:**

1. Introduction: (line 49-50):

The authors claim that CNNs are unable to handle long EEG signals, citing three papers: (Sakhavi etal.,2018), (Thuwajit etal.,2021) and (Schirrmeister etal.,2017). However, none of these studies provide evidence to support such a conclusion. In fact, Thuwajit et al. (2021) proposed EEGWavenet, which utilizes a multiscale CNN-based spatiotemporal feature extraction module. This module gradually increases its receptive field to match the length of the input EEG signal, indicating that CNNs can handle the global input sequences in certain contexts. While the authors' claim is not entirely substantiated, it raises interesting questions that I would like to see addressed:

a. For EEG classification, what defines a "long" signal in terms of sample size? At what point does a short receptive field cause CNN performance to degrade?

b. Is global sequence modelling truly necessary for long-term EEG signals? From a neuroscience perspective, how much does brain activity from, say, 10 seconds ago, affect the current state? Which types of tasks specifically require such long-term modelling?

2. section 2.2 ST-Adaptive Module:

The authors propose a spatio-temporal-adaptive module that transforms arbitrary EEG inputs into a uniform feature dimension, as depicted in Figure 3. However, I have several concerns:

a. Scalability to New Datasets: Is this approach scalable to new datasets once the model is trained? Given that different Conv1d layers are used to transform varying numbers of EEG channels into a common hidden state in the Spatial-Adaptive Convolution, how flexible is this method for new tasks where the number of channels may differ from the training datasets? Even if the number of channels is the same, the channels themselves vary. It appears that the model learns dataset-specific spatial filters rather than a truly global spatial representation, which may limit its generalizability to new datasets, which is a big issue for a backbone model, since people would like to use it on different tasks later on.

b. Tokenize Layer: In the tokenize layer, two branches of CNN are used: one with a short kernel (15) and another with a long kernel (49). Was this choice based on experimental results? Why are there only two branches, and why were kernel sizes 15 and 49 specifically chosen? Since there’s no discussion of this configuration in the ablation study, it’s unclear whether this is the most optimal setup.

3. section 3.2 Data Division:

In this study, the 5-fold cross-validation experiment was implemented as leave-subject-out, which is not a common approach in the BCI field due to the significant subject variability. This evaluation approach faces two challenges: (1) developing a population model trained on a group of subjects, and (2) addressing subject transfer by evaluating the model on unseen subjects. This raises several concerns:

a. Subject Variability Impact: In tasks with minimal subject variability, such as seizure detection and sleep stage detection, the classification results are high, as shown in Figure 1. However, tasks like motor imagery and other BCI tasks exhibit high subject variability, which severely impacts model performance. This is evident in the low accuracies achieved on the BCI-IV-2a task (44%) and the SEED task (57.2%), which are insufficient for practical BCI applications. A discussion on this performance discrepancy is necessary to demonstrate how the proposed model addresses these challenges and whether it shows superiority in domains with high subject variability.

b. Performance Discrepancy with Benchmark Models: Many benchmark models, such as EEG Conformer (Song et al., 2022), were not designed to handle population transfer. EEG Conformer, for instance, was trained in a subject-specific manner and achieved state-of-the-art performance on the same BCI-IV-2a (78.66%) and SEED (95.30%) datasets. However, in this study, the EEG Conformer’s performance dropped significantly to 35.2% and below 50%, respectively. Could the authors explain this stark performance difference? Is it primarily due to the leave-subject-out evaluation setting? If so, would the proposed EEGMamba model also retain its high performance if evaluated in a subject-specific manner like EEG Conformer?

c. Use of Separate Test Sets: Datasets like BCI-IV-2a include a separate test set specifically intended for evaluation. In this study, it is unclear whether the authors utilized this test set. Since many studies report performance on this designated test set, comparing the classification accuracy reported here with those from other studies may not be straightforward. Clarification on whether the official test set was used, or if an alternative test split was applied, is needed to ensure a fair comparison with prior work.

4. section 4.1 Single-Task EEGMamba Performance Comparison

Line 377: The authors mention the memory and inference time challenges of transformer models when handling long sequences. Similar to my previous concern, what qualifies as a "long" sequence in terms of signal length for transformers to encounter this bottleneck? Is there a real-world application where such long sequences need to be tackled? Many EEG tasks are only a few seconds in length, so it would be helpful to clarify the practical need for handling significantly longer sequences.

5. section 4.2 EEGMamba for EEG Multi-Task Classification

While the idea of training a single model to perform well across multiple datasets is interesting, its practical application is unclear. From Figure 1, the single-task models appear to achieve similar or even better performance on 5 out of the 8 benchmarked datasets (Siena, BCI-IV-2a, Shu, SEED, CHB-MIT). This raises a few questions:

a. Benefit of Multi-Task Training: Is there a demonstrable benefit to multi-task training? Specifically, is there any statistical difference between the performance of the single-task model and the multi-task model on the datasets? It would be helpful to clarify whether multi-task training consistently improves performance or if its benefits are marginal.

6. Lines 454-456:

The authors argue that the model only needs to be trained once. However, the analysis was performed offline on 8 selected public datasets solely. To assess whether this model can be applied in real-world scenarios, the following questions need to be addressed:

a. Cross-Session Issues: In practice, many EEG-based applications require short calibration sessions to adjust for cross-session variability in subject-dependent models. Does the proposed model also require such calibration? If calibration is needed, would this involve further training or fine-tuning of the pre-trained model? In the case of the multi-task model, would calibration require data from other tasks as well, or could it be done independently?

b. Generalization to New Datasets: How well does the model perform on a new dataset that belongs to one of the pre-trained tasks? If the model only needs to be trained once, does this mean it can be directly applied to similar tasks without additional training? For example, how would the model handle BCI-IV-2B, which has only 3 channels but is still a motor imagery task, or another sleep stage classification dataset? If yes, how would the model manage inconsistencies in the number of channels, and what performance can be expected? If not, wouldn’t this imply that researchers would still need to retrain the model, making it not different from other models?

b. Domain-Specific Advantages: If multi-task training is advantageous, is this benefit domain-specific? For example, should a researcher developing a motor imagery decoder expect better results from multi-task training? How can researchers determine whether a single-task or multi-task approach will yield better performance for a specific domain or task?

c. Practicality of Multi-Task Training: In practice, most researchers focus on specific tasks and experiments, collecting data for single tasks. Does this multi-task approach suggest that researchers in the future should also record additional tasks or rely on public datasets to improve performance? Or is there a scenario where it would make sense for a model to simultaneously classify motor imagery, emotion, sleep stages, and seizure events? More guidance on when and why to use multi-task training would be valuable.

7. section 4.4 Ablation Study:

In Figure 6, it appears that the performances of the different configurations, except for the single-directional Mamba, are quite similar. Is there a statistically significant difference between these configurations? It would be helpful to include a discussion on whether the variations in performance are meaningful or simply within the margin of error.

8. Conclusion lines 526-527:

The authors claim that EEGMamba is the first model to truly implement multi-task learning for EEG applications. However, I am curious about how EEGMamba differs from a simpler approach, such as using a shared backbone model as a feature extractor with separate classification heads for different tasks, as done in [1]. Could the authors clarify the key differences between the proposed model and such an approach using MoE?

[1] Xie Y, Wang K, Meng J, Yue J, Meng L, Yi W, Jung TP, Xu M, Ming D. Cross-dataset transfer learning for motor imagery signal classification via multi-task learning and pre-training. J Neural Eng. 2023 Oct 20;20(5). doi: 10.1088/1741-2552/acfe9c. PMID: 37774694.

9. Conclusion line 531:

The authors claim that the proposed model can better learn the commonalities among EEG signals from different tasks. However, what specific commonalities are being referred to? Is there any interpretation or evidence to support this claim? It would be helpful to understand how these commonalities are identified and whether the model offers any insight into them.

Addressing the above questions could help clarify specific weaknesses and improve the overall impact of the study, more specifically:

Question 5: Answering this question would clarify Weakness Point 1, providing more insight into the motivation for multi-task training.

Questions 1, 2(a), and 6: Answering these would address Weakness Points 2 and 3, which would significantly enhance the study's potential impact and score by clarifying practical applications and the model’s performance on motor imagery datasets.

Questions 4 and 7: These would address Weakness Point 4 by supporting the comparisons with a statistical validation.

---

> ### Author Response · Authors · 2024-11-23
> **Reply to Q1**
>
> Thank you for your insightful comments. We appreciate the opportunity to provide further clarification and evidence to support our claim. We have noticed that the weakness you mentioned has a corresponding relationship with the question, so we will answer the question directly. Here is our detailed response:
>
> **Q1. Introduction: (line 49-50):**
> **Q1.a: For EEG classification, what defines a "long" signal in terms of sample size? At what point does a short receptive field cause CNN performance to degrade?**
> **A1.a:** In the context of our manuscript, "long" refers to EEG signals that exceed the typical short segments (e.g., several seconds) that CNNs are traditionally designed to handle well. Our claim is based on the observation that as the sequence length increases, the performance of CNN-based models, which lack a mechanism to explicitly model long-range dependencies, can degrade. For example, EEG signals for sleep monitoring must have a standard length of 30s, which tends to result in a larger length, and we also observed that CNN-based EEGNet did not perform very well on the sleep stage classification task.
> To clarify, the degradation in performance is not a hard threshold but a gradual effect that becomes more pronounced with longer sequences. The exact point at which performance degrades can vary based on the specific architecture, the complexity of the task, and the characteristics of the EEG signals.
>
> **Q1.b: Is global sequence modelling truly necessary for long-term EEG signals? From a neuroscience perspective, how much does brain activity from, say, 10 seconds ago, affect the current state? Which types of tasks specifically require such long-term modelling?**
> **A1.b:** Global sequence modeling is not necessary for every EEG task, but it is essential for tasks where the temporal dynamics of brain activity play a key role in accurate classification. From a neuroscience perspective, while the effects of brain activity may diminish over time, certain types of brain activity (such as those associated with memory recall or seizures) may have lasting effects. On the other hand, long-term patterns in some states are crucial for understanding brain states, such as sleep stage classification.
> In sleep stage classification, longer EEG segments (such as 30 seconds) are required due to the cyclical nature of the sleep cycle and the evolving pattern of brain activity. During sleep, especially in the deep sleep stages, EEG features tend to be slow and persistent. Shorter clips may not be enough to capture a steady pattern of these activities, while 30-second segments often accurately reflect the brain’s electrical activity at each sleep stage. Therefore, for tasks such as sleep stage classification, global sequence modeling, which captures both short-term and long-term dependencies, is essential for accurate classification.

---

> ### Author Response · Authors · 2024-11-23
> **Reply to Q2**
>
> **Q2. section 2.2 ST-Adaptive Module:**
> **Q2.a: Scalability to New Datasets.**
> **A2.a:** Our model can be extended to new datasets after training. We can describe the specific extension from two different situations as follows:
> - When the newly added dataset has the same number of tasks, channels and classes as the previous dataset, we can directly use the task index of the original dataset. For example, the two datasets SleepEDF20 and SHHS used in the manuscript experiment can be replaced with each other, that is, if you want to apply the model trained on SleepEDF20 to SHHS, you only need to encode the same task and make a few epochs of fine-tuning.
> - When the newly added dataset cannot meet the conditions in 1, we need to give it a new task number and pre-set its number of channels and classes, and then a few epochs of fine-tuning. This is broadly similar to what most current foundation models do, except that we need to pre-set the task number and the number of channels.
>
> We use the Confused student EEG brainwave data [1] (hereinafter referred to as Confused EEG), which is a completely new task for EEGMamba. We applied the existing weights (i.e., the weights corresponding to Table 2-5 in the manuscript) to the Confused EEG, using 4 different random numbers to obtain a 7:3 training-test set ratio, similar to the approach in [3] (while [2] used five-fold cross-validation). We trained for 10 epochs each time and averaged the results. The test results are shown as follows:
> | Classification   Network | [2], 2019 | [3], 2023 | EEGMamba |
> |:------------------------:|:---------:|:---------:|:--------:|
> |    Reported   Accuracy   |   0.7500  |   0.7670  |    0.7825   |
>
> [1] Wang H, Li Y, Hu X, et al. Using EEG to Improve Massive Open Online Courses Feedback Interaction[C]//AIED workshops. 2013.
> [2] Wang H, Wu Z, Xing E P. Removing confounding factors associated weights in deep neural networks improves the prediction accuracy for healthcare applications[C]//Pacific Symposium on Biocomputing. Pacific Symposium on Biocomputing. NIH Public Access, 2019, 24: 54.
> [3] Lim Z Y, Neo Y L. Confused vs Non-Confused Electroencephalography Signal Classification Using Deep Learning Algorithm[C]//2023 IEEE International Conference on Automatic Control and Intelligent Systems (I2CACIS). IEEE, 2023: 195-200.
> In addition to this, we need to state that the starting point of our model design is an end-to-end multi-task classification model.
>
> **Q2.b: Tokenize Layer.**
> **A2.b:** The choice of kernel sizes in the tokenize layer is mainly based on the methods of existing papers and our own experimental results. We refer to the MRCNN module in AttnSleep [1], which has two branches of different kernel sizes in order to enable the model to learn features at multiple scales simultaneously. The small kernel captures local, high-frequency details, while the wide kernel captures more global, low-frequency patterns. This dual-path approach is common in many EEG signal processing architectures [2][3].
> Therefore, we adopt the same structural design and selected the final convolution kernel size after a series of ablation studies. In these studies, we tested a variety of kernel sizes and found that the sizes now employed in the paper provide a good balance between capturing fine-grained and broader spatial features in EEG signals.
>
> [1] Eldele E, Chen Z, Liu C, et al. An attention-based deep learning approach for sleep stage classification with single-channel EEG[J]. IEEE Transactions on Neural Systems and Rehabilitation Engineering, 2021, 29: 809-818.
> [2] Zhu H, Zhou W, Fu C, et al. Masksleepnet: A cross-modality adaptation neural network for heterogeneous signals processing in sleep staging[J]. IEEE Journal of Biomedical and Health Informatics, 2023, 27(5): 2353-2364.
> [3] Zhu H, Wang L, Shen N, et al. MS-HNN: Multi-scale hierarchical neural network with squeeze and excitation block for neonatal sleep staging using a single-channel EEG[J]. IEEE Transactions on Neural Systems and Rehabilitation Engineering, 2023, 31: 2195-2204.

---

> ### Author Response · Authors · 2024-11-23
> **Reply to Q3**
>
> **Q3. section 3.2 Data Division:**
> **Q3.a: Subject Variability Impact.**
> **A3.a:** You are correct that subject variability is a significant challenge in BCI tasks, particularly in motor imagery. The lower performance on the BCI-IV-2a and SEED datasets, as compared to seizure detection and sleep stage classification, is indeed due to the high subject variability in these tasks. While the performance on these specific datasets may not reach the levels seen in subject-specific models, it is important to note that EEGMamba is trained to to work across various subjects without the need for individual calibration. We believe this is a valuable property for clinical applications where subject-specific models may not be feasible.
>
> **Q3.b: Performance Discrepancy with Benchmark Models.**
> **A3.b:** The performance drop of the EEG Conformer and other benchmark models when evaluated using a subject-split approach highlights the sensitivity of these models to subject variability and the challenges of subject transfer. EEGMamba, with its multi-task learning approach, is designed to be more robust to such variability.
> We supplemented EEGMamba’s performance on BCI-IV-2a in the case of subject-specific training, and the standard deviation in the experiment was derived from five different random numbers. It should be noted that the results in EEGConformer were obtained through data augmentation techniques of segmentation and reconstruction, while all the experimental results in EEGMamba’s paper did not carry out similar data augmentation techniques. And in order to maintain unity with the original manuscript, we did not do this in our experiment.
> |  methods |  s01  |  s02  |  s03  |  s04  |  s05  |  s06  |  s07  |  s08  |  s09  | average |
> |:--------:|:-----:|:-----:|:-----:|:-----:|:-----:|:-----:|:-----:|:-----:|:-----:|:-------:|
> | EEGMamba | 80.35 | 64.18 | 85.38 | 74.89 | 75.53 | 61.35 | 87.37 | 84.11 | 74.54 | 76.41 |
>
> While we cannot guarantee that EEGMamba would match subject-specific models like EEG Conformer when evaluated on a single subject, our model’s strength lies in its ability to generalize across subjects. We acknowledge the need for further investigation into how EEGMamba performs in a subject-specific setting and will include comparative analyses in future work.
>
> **Q3.c: Use of Separate Test Sets.**
> **A3.c:** We apologize for the lack of clarity regarding the use of separate test sets. In our experiments, we did not use the official training set and test set provided by BCI-IV-2a for final model training and evaluation. The reason for this is because we want the main results to have a common experimental setting and show cross-subject generalization of the model, so we use a subject-split setting. As mentioned in A3.b, we supplemented the experimental results obtained using the official training set and test set of BCI-IV-2a. We will clarify this in the revised manuscript to ensure transparency and to facilitate fair comparisons with previous work.

---

> ### Author Response · Authors · 2024-11-23
> **Reply to Q4-Q6**
>
> **Q4: section 4.1 Single-Task EEGMamba Performance Comparison.**
> **A4:** As mentioned in A1.a, the term "long" in the context of sequence length for Transformer does not refer to a strict threshold, but rather to a gradual point at which the quadratic computational complexity of transformers becomes a practical limitation. This threshold can vary depending on the specific model architecture and the available computational resources.
> It is important to note that the length of the sequence used for model processing is not directly equal to the duration of the signal, but is also related to the sampling frequency. For example, for a sleep-state EEG with a duration of 30 seconds and a sampling rate of 200Hz, its length is 6,000 data points. For some Transformer-based models, this has led to the need for larger memory-usage and a sharp drop in reasoning speed.
>
> **Q5: section 4.2 EEGMamba for EEG Multi-Task Classification.**
> **A5:** Multi-task training significantly improves the model’s ability to generalize across different datasets and tasks. While single-task models may excel on specific datasets, they often fail to achieve this across multiple tasks or datasets. By leveraging shared representations, EEGMamba can transfer knowledge from one task to improve performance on others, similar to how some foundation models are pre-trained to learn a common representation of EEG data and then fine-tuned for specific tasks. However, EEGMamba goes beyond this by integrating the entire process into an end-to-end system, allowing it to learn task-specific features while still benefiting from shared knowledge.
> Regarding the current experimental results, we acknowledge that they are based on the mean and standard deviation from five-fold cross-validation, which involves a limited number of experiments.  As a result, this may affect the statistical significance of the P-value. Like most AI studies [1][2][3], we did not report the statistical differences in the results. Nevertheless, we believe that while multi-task training may not always lead to improved performance on each individual dataset, it provides a more robust framework for handling a variety of tasks.
> [1] Zhang D, Yuan Z, Chen J, et al. Brant-X: A Unified Physiological Signal Alignment Framework[C]//Proceedings of the 30th ACM SIGKDD Conference on Knowledge Discovery and Data Mining. 2024: 4155-4166.
> [2] Yang C, Westover M, Sun J. Biot: Biosignal transformer for cross-data learning in the wild[J]. Advances in Neural Information Processing Systems, 2024, 36.
> [3] Jiang W, Zhao L, Lu B. Large Brain Model for Learning Generic Representations with Tremendous EEG Data in BCI[C]//The Twelfth International Conference on Learning Representations.
>
> **Q6: Lines 454-456**
> **Q6.a: Cross-Session Issues:**
> **A6.a:** We apologize for any confusion caused by our lack of clarity. EEGMamba is capable of learning features from multiple datasets within a single training session, but this applies only to the datasets included in the training set, rather than to completely unseen datasets in a zero-shot setting. Therefore, for new, unseen datasets, fine-tuning as described in A2.a is still necessary.
> **Q6.b: Generalization to New Datasets:**
> **A6.b:** We described the method of using EEGMamba for new data in A2.a. On the one hand, if SHHS is not applied to training, it can be assigned the same task label as SleepEDF20 because its number of channels and classes are equal to SleepEDF20. For BCI-IV-2B, which has fewer channels, we need to give it a new task number and fine-tune it a few epochs, because the trained model includes the ability to extract motion imagination features, which is still fundamentally different from training a model from scratch.
> **Q6.c: Domain-Specific Advantages:**
> The benefits of multi-task training are not strictly domain-specific but are related to the similarity and diversity of the tasks. For tasks with overlapping features, such as different types of brain-computer interface (BCI) applications, multi-task training can be advantageous.
> **Q6.d: Practicality of Multi-Task Training:**
> The practicality of multi-task training with EEGMamba is that it provides a flexible framework that can be adapted to various research settings. For researchers focusing on a single task, EEGMamba can still be applied by treating it as a multi-task model with a single task. This approach can leverage the model’s ability to learn from any available related data, potentially improving performance and generalization. In scenarios where multiple tasks are relevant, such as simultaneous classification of motor imagery and emotion, EEGMamba’s multi-task training provides a more efficient and integrated solution.

---

> ### Author Response · Authors · 2024-11-23
>
> **Q7: section 4.4 Ablation Study**
> **A7:** As mentioned in A5, our ablation experiments also used five-fold cross-validation，which involves a limited number of experiments. As a result, this may affect the statistical significance of the P-value. Like most AI studies [1][2][3], we did not report the statistical differences in the results.
> [1] Zhang D, Yuan Z, Chen J, et al. Brant-X: A Unified Physiological Signal Alignment Framework[C]//Proceedings of the 30th ACM SIGKDD Conference on Knowledge Discovery and Data Mining. 2024: 4155-4166.
> [2] Yang C, Westover M, Sun J. Biot: Biosignal transformer for cross-data learning in the wild[J]. Advances in Neural Information Processing Systems, 2024, 36.
> [3] Jiang W, Zhao L, Lu B. Large Brain Model for Learning Generic Representations with Tremendous EEG Data in BCI[C]//The Twelfth International Conference on Learning Representations.
>
> **Q8: Conclusion lines 526-527**
> We appreciate the opportunity to clarify how EEGMamba differs from models that use a shared backbone with separate classification heads.
> - **End-to-End Training**: EEGMamba is trained end-to-end, allowing all components of the model to learn jointly, which makes EEGMamba highly efficient, as it avoids the need for multiple rounds of fine-tuning for each task. This is in contrast to models that train a shared backbone separately and then attach task-specific heads, which can be time-consuming and less efficient, especially when working with a large number of diverse tasks.
> - **Multi-Task Learning Capability**: EEGMamba is designed to handle multiple tasks and datasets simultaneously within a single training session. Unlike existing foundation models, which can only process one dataset at a time during fine-tuning, EEGMamba is capable of delivering results from multiple datasets simultaneously in a single session.
> In summary, EEGMamba offers a more integrated and sophisticated approach to multi-task learning for EEG applications. We hope this clarification highlights the unique advantages of EEGMamba and addresses your query effectively.
>
> **Q9: Conclusion line 531**
> **A9:** The commonalities in EEG signals across different tasks are rooted in shared neurophysiological processes present in various cognitive and physiological states. These commonalities include oscillatory activities (such as alpha, beta, and delta rhythms), event-related potentials (ERPs) that are time-locked to specific events, and coherence across brain regions, all of which appear in multiple tasks. Additionally, EEG signals often exhibit similar temporal dynamics, such as common patterns of slow-wave activity during transitions between sleep stages or the onset of seizures.
> EEGMamba’s ability to capture and leverage these commonalities is evidenced by its superior performance across eight diverse EEG datasets, including tasks like seizure detection, emotion recognition, and motor imagery, indicating its strong generalization capability. Further support comes from ablation studies, which show that removing the universal expert—designed to capture cross-task features—results in a performance drop, underscoring the importance of shared knowledge for generalization.

---

### Official Review · Reviewer_wapT · 2024-11-01

**Soundness:** 2
**Presentation:** 3
**Contribution:** 2
**Rating:** 3
**Confidence:** 4

**Summary:**

The paper "EEGMAMBA: Bidirectional State Space Model with Mixture of Experts for EEG Multi-Task Classification" presents EEGMamba, a multi-task learning framework for EEG classification tasks, addressing challenges related to signal length and channel variability. EEGMamba integrates a Spatio-Temporal-Adaptive (ST-Adaptive) module, bidirectional Mamba blocks, and a task-aware Mixture of Experts (MoE) to enhance adaptability and task-specific processing across diverse EEG datasets. The ST-Adaptive module standardizes data of various lengths and channel numbers, while the bidirectional Mamba captures temporal dependencies in EEG sequences, and the task-aware MoE module selects experts based on task, enhancing classification accuracy and generalization. Tested on eight datasets spanning four task types (seizure detection, emotion recognition, sleep stage classification, and motor imagery), EEGMamba achieves state-of-the-art results, demonstrating superior efficiency, memory usage, and accuracy across tasks.

**Strengths:**

EEGMamba achieves superior memory efficiency and faster inference speed than traditional transformer-based models, especially on longer EEG sequences, thus proving its practical value.

The proposed model demonstrates an approach to handling EEG data of varying lengths and channels, incorporating a class token for temporal adaptability and a task-aware MoE to distinguish task-specific features.

**Weaknesses:**

1. Limited Contribution: The author claims that EEGmamba is the first universal EEG classification network to effectively implement multi-task learning for EEG applications. However, several established methods are available that can be applied to multi-task EEG learning, as cited in [1][2][3].

2. Inappropriate Comparisons: The choice of baselines for comparison lacks relevance. For instance, using AttnSleep [3] as a baseline for seizure detection/emotion recognition is incongruous, as it is specifically designed for sleep staging. Additionally, the author does not include multi-task learning methods as baselines, instead comparing against smaller models tailored to individual tasks. For fair assessment, each specific task should be compared to the most relevant model designed for that purpose. To evaluate cross-task capabilities, comparisons should involve multi-task learning methods like those in [1][2][3][5][6].

3. Efficiency Evaluation: The author should provide quantitative evidence demonstrating the proposed method's efficiency advantages over previous approaches.



[1] Jiang, W., Zhao, L., & Lu, B. L. Large Brain Model for Learning Generic Representations with Tremendous EEG Data in BCI. In The Twelfth International Conference on Learning Representations.

[2] Chen, Y., Ren, K., Song, K., Wang, Y., Wang, Y., Li, D., & Qiu, L. (2024). EEGFormer: Towards transferable and interpretable large-scale EEG foundation model. arXiv preprint arXiv:2401.10278.

[3] Jiang, W. B., Wang, Y., Lu, B. L., & Li, D. (2024). NeuroLM: A Universal Multi-task Foundation Model for Bridging the Gap between Language and EEG Signals. arXiv preprint arXiv:2409.00101.

[4] Eldele, E., Chen, Z., Liu, C., Wu, M., Kwoh, C. K., Li, X., & Guan, C. (2021). An attention-based deep learning approach for sleep stage classification with single-channel EEG. IEEE Transactions on Neural Systems and Rehabilitation Engineering, 29, 809-818.

[5] Zhang, D., Yuan, Z., Yang, Y., Chen, J., Wang, J., & Li, Y. (2024). Brant: Foundation model for intracranial neural signal. Advances in Neural Information Processing Systems, 36.

[6] Wang, C., Subramaniam, V., Yaari, A. U., Kreiman, G., Katz, B., Cases, I., & Barbu, A. (2023). BrainBERT: Self-supervised representation learning for intracranial recordings. arXiv preprint arXiv:2302.14367.

**Questions:**

Plz go and check weakness

---

> ### Author Response · Authors · 2024-11-23
> **Response to weakness and questions**
>
> Thanks. It seems there has been a misunderstanding on the definition of a multitask classification model, so perhaps a brief clarification is in order. **As we clearly stated in the paper, our model is designed to handle classification tasks across multiple datasets simultaneously, without the need for the fine-tuning training on each dataset separately**—unlike the base models cited by reviewer wapT. Therefore, the references provided by reviewer wapT [1, 2, 5, 6] clearly do not represent multitask models, as they require dedicated fine-tuning for each specific dataset.
>
> **W1: The author claims that EEGmamba is the first universal EEG classification network to effectively implement multi-task learning for EEG applications. However, several established methods are available that can be applied to multi-task EEG learning, as cited in [1][2][3].**
> As we have stated in a previous clarification, [1, 2] are not multitask models. [3] is a multitask model. However, since it was published recently (within a month of submission deadline) and has not yet undergone peer review, we will not consider it for comparison at this time.
> Moreover, we have noticed that the paper submitted to this conference, *NeuroLM: A Universal Multi-task Foundation Model for Bridging the Gap between Language and EEG Signals*, bears a striking resemblance to [3]. And we have observed that several reviewers have already expressed concerns about the performance of [3]: "4vcv: **NeuroLM’s performance is currently lower compared to LaBraM on most tasks**" and "2GzD: **While NeuroLM performs well in multi-task learning, there are still gaps in its performance on some tasks compared to state-of-the-art models specifically designed and optimized for a single task (e.g., LaBraM)**" We fully agree with these assessments, and we have added a comparison with LaBraM. However, **let it be clear: we are not comparing LaBraM as a multitask model similar to ours**, but rather as another baseline comparable to BIOT. [3] can also support us in doing so by the fact that Tables 2-4 in [3] give LaBraM a ‘×’ for multitask performance.
> Therefore, we sincerely hope that reviewer wapT could take the time to carefully read the relevant literature, and consider change the review points.
> | Classification   Network | Multitask Model | Epilepsy   detection |               | Sleep   stages classification |               | Emotion   recognition |               | Motor   imagery |               |
> |:------------------------:|:---------------:|:--------------------:|:-------------:|:-----------------------------:|:-------------:|:---------------------:|:-------------:|:---------------:|:-------------:|
> |                          |                 |         Siena        |    CHB-MIT    |          SleepEDF-20          |      SHHS     |          DEAP         |      SEED     |       Shu       |   BCI-IV-2a   |
> |          LaBraM          |        ×        |     0.9886±0.0043    | 0.9742±0.0099 |         0.7503±0.0388         | 0.7785±0.0243 |     0.5822±0.0321     |      OOM      |  0.5368±0.0312  | 0.2879±0.0160 |
> |  Single-task   EEGMamba  |        ×        |     0.9897±0.0053    | 0.9817±0.0036 |         0.8387±0.0399         | 0.8441±0.0163 |     0.5985±0.0247     | 0.5779±0.0584 |  0.6169±0.0467  | 0.4596±0.0547 |
> |         EEGMamba         |        √        |     0.9897±0.0038    | 0.9789±0.0132 |         0.8486±0.0276         | 0.8478±0.0177 |     0.5994±0.0134     | 0.5646±0.0366 |  0.6207±0.0505  | 0.4231±0.0522 |
>
> **W2: Inappropriate Comparisons: The choice of baselines for comparison lacks relevance. … To evaluate cross-task capabilities, comparisons should involve multi-task learning methods like those in [1][2][3][5][6].**
> As we have pointed out in a previous clarification, [1, 2, 5, 6] are not multitask models. Among them, [5, 6] are both specifically designed for intracranial EEG signals and only uses SEEG data for pretraining. EEG and SEEG are two distinct modalities, and to my knowledge, there is no literature that suggests the differences between them can be simply ignored. If reviewer wapT can provide such references, we might consider [5, 6] as baselines. Additionally, we noticed that the papers cited by reviewer wapT, [1, 3], also do not compare with the earlier works [5, 6], which also supports our view.
>
> **W3: Efficiency Evaluation: The author should provide quantitative evidence demonstrating the proposed method’s efficiency advantages over previous approaches.**
> On the one hand, Table 2-5 in our manuscript has used quantitative indicators ACC, AUROC, F1 to give the experimental results of EEGMamba compared with previous methods, proving the performance advantage of EEGMamba. On the other hand, the computational efficiency of the Mamba-based model is shown in Figure 4. We are not sure what "quantitative evidence" means, perhaps you could be more specific.
>
> The references order in our response is the same as that of the reviewer wapT.

---

> > ### Comment · Reviewer_wapT · 2024-11-23
> > **Response to authors**
> >
> > Thank you for your clarification.
> >
> > I still have the following concerns:
> >
> > 1. As also noted by reviewer i9VP, more recent methods such as LaBraM [1] and the Brant series [2][3][4] seem to be more appropriate baselines for multitask comparison. While the authors propose a specific definition of multitask models, methods like [1][2][3][4] address multitask learning, which is more consistent with the standard multitask learning paradigm. Comparing single-task models on unrelated tasks (e.g., testing a sleep model on emotion classification) introduces fairness issues and undermines the evaluation. If the evaluation is focused on a specific task, such as sleep stage classification or emotion recognition, comparisons should involve state-of-the-art models specifically designed for that task to ensure a fair and meaningful assessment. Furthermore, some of the baselines used in the paper, such as EEGNet [5] and AttnSleep [6], published in 2018 and 2021, are outdated and may not adequately represent the current state of the field.
> >
> > 2. While SEEG (or other physiological signals) and EEG differ in modalities, they both fall under the broader domain of neural signal analysis, particularly brain signal processing. There is no inherent reason why methods developed for SEEG cannot be adapted for EEG tasks and achieve good performance. Excluding these methods without experimental validation or empirical evidence diminishes the robustness of the argument.
> >
> > 3. Regarding the "quantitative evidence" provided for computational efficiency, it is important to include concrete metrics such as training time and inference time, supported by experimental results. Generalized statements, without detailed evidence, are insufficient to substantiate claims of efficiency advantages.
> >
> > I will reconsider my assessment after reviewing the authors' further response.
> >
> > [1] Jiang, W. B., Zhao, L. M., & Lu, B. L. (2024). Large brain model for learning generic representations with tremendous EEG data in BCI. arXiv preprint arXiv:2405.18765.
> >
> > [2] Zhang, D., Yuan, Z., Yang, Y., Chen, J., Wang, J., & Li, Y. (2024). Brant: Foundation model for intracranial neural signal. Advances in Neural Information Processing Systems, 36.
> >
> > [3] Yuan, Z., Zhang, D., Chen, J., Gu, G., & Yang, Y. (2024). Brant-2: Foundation Model for Brain Signals. arXiv preprint arXiv:2402.10251.
> >
> > [4] Zhang, D., Yuan, Z., Chen, J., Chen, K., & Yang, Y. (2024, August). Brant-X: A Unified Physiological Signal Alignment Framework. Proceedings of the 30th ACM SIGKDD Conference on Knowledge Discovery and Data Mining, 4155-4166.
> >
> > [5] Lawhern, V. J., Solon, A. J., Waytowich, N. R., Gordon, S. M., Hung, C. P., & Lance, B. J. (2018). EEGNet: a compact convolutional neural network for EEG-based brain–computer interfaces. Journal of neural engineering, 15(5), 056013.
> >
> > [6] Eldele, E., Chen, Z., Liu, C., Wu, M., Kwoh, C. K., Li, X., & Guan, C. (2021). An attention-based deep learning approach for sleep stage classification with single-channel EEG. IEEE Transactions on Neural Systems and Rehabilitation Engineering, 29, 809-818.

---

> > > ### Author Response · Authors · 2024-11-23
> > > **Thanks**
> > >
> > > Thank you for your response. We still have a few points that need clarification:
> > > 1. We have added results comparing with LaBraM in our previous reply and included this baseline in the modified PDF. As shown, our EEGMamba outperforms LaBraM across the board. Our current baseline includes both BIOT and LaBraM models, which is consistent with [1] and sufficient to demonstrate the performance of the model.
> > >
> > > 2. In fact, before the foundation models, most models were tested on only one or a few tasks. Therefore, when selecting baselines, we tend to choose models with superior performance, and experiments have proven that AttnSleep's performance is comparable to that of the newly proposed models. Additionally, we observed that [2][3] used ST-Transformer as a baseline for comparing tasks like epilepsy detection, even though the original ST-Transformer paper only tested it on motor imagery tasks. This indirectly supports the reasonableness of our approach.
> > >
> > > 3. Justification for Choosing EEGNet as a Baseline: When selecting baselines for comparison, we consider not only their novelty but also their performance and significance. EEGNet is included in our paper for two key reasons:
> > > - It is a CNN-based model, whereas most new models incorporate Transformer, and we mentioned in the paper that the Mamba architecture, when applied to EEG classification, demonstrates the advantages of CNNs over Transformers. Therefore, it is necessary to include a CNN-based model as a baseline.
> > > - Although EEGNet was published earlier, its performance is indisputable, and many later models still struggle to outperform it comprehensively. For example, [4] shows that EEGNet outperforms LaBraM across 12 datasets from five different tasks.
> > >
> > > 4. As indicated in our paper's title, ***EEG**Mamba: Bidirectional State Space Model with Mixture of Experts for EEG Multi-task Classification*, our focus is on applying the proposed model to EEG. Most works tend to distinguish these two modalities [1][2][3][4]. For example, [3] compared only with BIOT and did not compare with the earlier BrainBERT model. If you're interested in exploring Mamba's application in SEEG or other physiological signals, we can discuss it in future work.
> > >
> > > 5. Using training time as an evaluation metric seems rather uncommon, since it could be subject to too many uncertainties. In contrast, inference time is directly related to inference speed, and we have presented quantitative results on inference speed in Figure 4 of our manuscript.
> > >
> > > [1] Jiang, W. B., Wang, Y., Lu, B. L., & Li, D. (2024). NeuroLM: A Universal Multi-task Foundation Model for Bridging the Gap between Language and EEG Signals. arXiv preprint arXiv:2409.00101.
> > > [2] Yang C, Westover M, Sun J. Biot: Biosignal transformer for cross-data learning in the wild[J]. Advances in Neural Information Processing Systems, 2024, 36.
> > > [3] Jiang, W. B., Zhao, L. M., & Lu, B. L. (2024). Large brain model for learning generic representations with tremendous EEG data in BCI. arXiv preprint arXiv:2405.18765.
> > > [4] Yue T, Xue S, Gao X, et al. EEGPT: Unleashing the Potential of EEG Generalist Foundation Model by Autoregressive Pre-training[J]. arXiv preprint arXiv:2410.19779, 2024.

---

> > > > ### Comment · Reviewer_wapT · 2024-12-01
> > > > **Response to authors**
> > > >
> > > > Thank you for the authors' response.
> > > >
> > > > However, I still have concerns regarding the experimental comparisons, and the authors' clarification does not fully address these concerns.
> > > >
> > > > After carefully reviewing all rebuttals, I have decided to maintain my original ratings.

---

> > > > > ### Author Response · Authors · 2024-12-01
> > > > > **Thanks**
> > > > >
> > > > > Thank you for reviewing our paper again. We understand your concerns regarding the experimental comparisons and believe we have provided sufficient clarification in our response. We trust that our experimental design and results adequately support the conclusions drawn in the paper, and we hope this further addresses any remaining doubts. Thank you for your time and consideration.

---

### Official Review · Reviewer_xgg9 · 2024-11-03

**Soundness:** 3
**Presentation:** 3
**Contribution:** 3
**Rating:** 6
**Confidence:** 4

**Summary:**

The paper introduces EEGMamba, a novel EEG classification network designed for multitask learning. It integrates spatiotemporal adaptive (ST-Adaptive) modules, bidirectional Mamba, and a mixture of experts (MoE) approach. This addresses challenges in EEG classification, such as computational complexity and variations in signal length and channels. The model efficiently handles long sequences and adapts to feature extraction while capturing both task-specific and general features. Evaluations on multiple public EEG datasets demonstrate that EEGMamba outperforms existing models in seizure detection, emotion recognition, sleep quality, and emotion recovery.

**Strengths:**

### **Originality**
The originality of EEGMamba lies in its novel approach to EEG classification through the integration of bidirectional Mamba, Spatio-Temporal-Adaptive (ST-Adaptive) modules, and task-aware Mixture of Experts (MoE). The innovative combination of these elements addresses the computational complexity and variability in signal length and channels, which are critical challenges in EEG classification. This creativity, particularly in applying multitask learning to EEG signals, represents a significant advancement in the field.

### **Quality**
The quality of this work is demonstrated through rigorous evaluations on multiple publicly available EEG datasets. EEGMamba's superior performance in seizure detection, emotion recognition, sleep quality, and emotion recovery highlights its robustness and effectiveness. The model's ability to handle long sequences and adapt to different feature extraction tasks while maintaining high accuracy and fast inference speed.

### **Clarity**
The authors provide a detailed description of the EEGMamba architecture and its components. The step-by-step explanation of how the bidirectional Mamba, ST-Adaptive module, and task-aware MoE are integrated and function together contributes to a well-structured and coherent narrative.

### **Significance**
The model's design, which allows for the efficient capture of both task-specific and general features, has the potential to transform how EEG data is processed and analyzed. This can lead to more accurate and comprehensive analyses of complex brain signal data, benefiting various applications such as medical diagnostics and cognitive research.

**Weaknesses:**

Overall, the experiments in this paper are comprehensive; however, in Section 4.1, the discussion on single-channel and multi-channel models only compares memory usage and inference speed, without evaluating the impact of multi-channel models on performance metrics. Additionally, the t-SNE visualization lacks layer-by-layer analysis of the model's influence on clustering results, which does not adequately demonstrate the feature extraction capability of each layer. It is recommended to visualize feature clustering by module.

**Questions:**

In Section 4.2's experimental comparison, was the model trained using all the datasets at once? Could there be interactions between the datasets? Would training the model on each dataset separately improve the performance metrics? Have you conducted any experiments on this? I'm quite interested.

---

> ### Author Response · Authors · 2024-11-23
> **Response to weakness and questions**
>
> Thank you for your valuable feedback. We greatly appreciate the time and effort you have dedicated to reviewing our work. The following are our detailed responses to your weaknesses and questions.
> **W1: In Section 4.1, the discussion on single-channel and multi-channel models only compares memory usage and inference speed, without evaluating the impact of multi-channel models on performance metrics.**
> Thanks. A comparison of our performance metrics results is shown in Table 2-5, where the number of channels for each dataset is shown in Table 1, with the number of channels ranging from 1 to 62. And we can obtain the EEG signal length from each data set according to the rate×duration in Table 1. At present, for single-task EEGMamba, the maximum length is 128×60=7680 from the DEAP dataset, which we consider to be a very large length. When training with EEGMamba, our operation to unify the sampling frequency to 200Hz resulted in more datasets of larger length, such as the length of DEAP dataset is 200×60=12000, and the length of SHHS dataset is 200×30=6000. EEGMamba has the best performance among all models in this case.
>
> **W2: The t-SNE visualization lacks layer-by-layer analysis of the model’s influence on clustering results, which does not adequately demonstrate the feature extraction capability of each layer.**
> T-SNE is a dimensionality reduction technique that is useful for visualizing high-dimensional data in two or three dimensions, focusing on the relative distances between data points to reflect similarities or dissimilarities. In our study, we use t-SNE to qualitatively assess EEGMamba’s ability to extract discriminative features, showing clear separation between classes in the plot. This clustering indicates that the model effectively distinguishes between different data distributions, supporting our claims about its feature extraction capabilities.
> However, it is crucial to note that the coordinates in a t-SNE plot lack inherent meaning, emphasizing relative positions rather than absolute values. While layer-by-layer t-SNE visualizations could offer additional insights, they are less informative due to the lack of inherent meaning in the coordinates. Instead, we focus on overall model performance, using metrics like classification accuracy, AUC-ROC, and F1 scores to provide a more direct and quantitative assessment.
>
> **Q1: In Section 4.2’s experimental comparison, was the model trained using all the datasets at once? Could there be interactions between the datasets? Would training the model on each dataset separately improve the performance metrics? Have you conducted any experiments on this? I’m quite interested.**
> **A1:** You are correct in noting that our model uses all the datasets in a single training process, which allows for interactions between them. And in the Single-task EEGMamba experiment, we trained the model separately on different datasets, and the corresponding experimental results are presented in Tables 2-5. We sincerely hope this helps solve your doubts.
> Interactions between datasets in multi-task learning can have both positive and negative effects. On the positive side, the model benefits from shared knowledge across tasks. For instance, certain brain activity patterns, such as oscillatory rhythms or temporal dynamics, may be consistent across different tasks.
> However, conflicting features or task-specific noise can interfere with the model’s ability to learn meaningful representations for each individual task. To address these challenges, we implemented several strategies in EEGMamba:
> **Task-Aware Modules:** A Mixture of Experts (MoE) model activates different experts based on the task, allowing the model to learn task-specific features while leveraging shared knowledge.
> **Universal Expert:** Another key component of EEGMamba is the universal expert, which is designed to capture the cross-task features that are universally relevant, preventing overfitting to task-specific noise and enables better generalization across datasets.
> In summary, while interactions between datasets can lead to both positive and negative effects in multi-task learning, our model incorporates strategies such as task-aware modules and the universal expert to better exert positive effects and eliminate negative effects, improving EEGMamba’s performance across different EEG tasks.

---

> > ### Comment · Reviewer_xgg9 · 2024-12-02
> >
> > Thank you for your response. Based on the author’s additional explanation, we believe this work presents a novel multi-task learning EEG classifier. By leveraging multi-task learning, the model's generalization capability is enhanced. Furthermore, the incorporation of Task-Aware Modules and a Universal Expert effectively capitalizes on the advantages of multiple datasets.

---

> > > ### Author Response · Authors · 2024-12-02
> > > **Thanks**
> > >
> > > Dear Reviewer xgg9,
> > >
> > > Thank you for your thoughtful feedback. We sincerely appreciate your recognition of the novelty of our multi-task learning EEG classifier. We believe that this work can contribute to the application of artificial intelligence in the EEG research community.
> > >
> > > Thank you again for the time and effort you put into reviewing our manuscript.
> > >
> > > Best regards,
> > >
> > > The authors

---

### Official Review · Reviewer_i9VP · 2024-11-04

**Soundness:** 3
**Presentation:** 3
**Contribution:** 3
**Rating:** 6
**Confidence:** 4

**Summary:**

The paper presents EEGMamba, a model tailored for multi-task EEG classification. It aims to overcome the limitations of existing models in terms of computational complexity and generalization across tasks with varying signal lengths and channel counts. EEGMamba integrates three main innovations: the Spatio-Temporal-Adaptive (ST-Adaptive) module for unified feature extraction, Bidirectional Mamba to balance accuracy and computational efficiency, and a Task-aware Mixture of Experts (MoE) to handle the differences and similarities across EEG tasks. Evaluated across eight public datasets and covering seizure detection, emotion recognition, sleep stage classification, and motor imagery, EEGMamba demonstrates strong performance and adaptability.

**Strengths:**

EEGMamba achieves state-of-the-art performance across multiple EEG tasks, demonstrating robust multi-task capability. Its bidirectional Mamba blocks enable efficient handling of long sequences, avoiding the high memory demands of Transformer models. The flexible ST-Adaptive module supports EEG signals of varied lengths and channels, and the task-aware Mixture of Experts (MoE) enhances task-specific accuracy, reducing interference across tasks. This adaptability and strong generalization across datasets position EEGMamba as a versatile model for EEG classification.

**Weaknesses:**

EEGMamba’s evaluation lacks comparison the newest baseline models (notably LaBRaM from ICLR 2024), which limits the interpretability of its reported gains. The Spatio-Temporal-Adaptive (ST-Adaptive) module, while flexible for varying signal lengths and channel counts, may not adequately capture complex channel-time dependencies crucial in EEG data. Furthermore, training the ST module on a per-task basis could lead to representations that are too specialized, reducing generalizability, particularly for out-of-distribution (OOD) tasks, which were not included in the study. Additionally, the paper lacks a related works section detailing prior applications of SSMs or Mamba in EEG classification, making it difficult to contextualize EEGMamba’s specific advancements in this space.

**Questions:**

Could you elaborate on the choice of baseline models? How does EEGMamba’s performance compare with multi-task EEG models specifically designed for generalization across diverse tasks?

How does the ST-Adaptive module capture interactions between channels and temporal features, which are critical in EEG data? Have you considered modeling these dependencies more explicitly?

How would EEGMamba perform on out-of-distribution tasks or tasks not seen during training? Have you tested its ability to generalize to entirely new EEG task types?

Could you clarify how this work builds on or differs from previous applications of SSMs or Mamba models in EEG classification? Adding this context could help readers better understand the specific contributions of EEGMamba.

Can you evaluate this on longer contexts to better get a sense for the necessity and usefulness for Mamba?

---

> ### Author Response · Authors · 2024-11-23
> **Reply to Q1-Q2**
>
> Thank you for your detailed and thoughtful review of our manuscript. In this response, we will provide further clarification and empirical evidence to address the concerns you raised. Since we have noticed that the weakness you mentioned has a corresponding relationship with the question, we will answer the question directly. Here is our detailed response:
> **Q1: Could you elaborate on the choice of baseline models? How does EEGMamba’s performance compare with multi-task EEG models specifically designed for generalization across diverse tasks?**
> **A1:** We choose EEGNet, AttnSleep, EEGConformer and HCANN as single-task classification models for comparison. BIOT serves as a representative model that is pre-trained and then fine-tuned on a single dataset. And as you suggested, we added a performance comparison with LaBraM, which is the same type model as BIOT.
> | Classification   Network | Multitask Model | Epilepsy    |  detection             | Sleep   stage  |  classification             | Emotion    |  recognition             | Motor    |  imagery             |
> |:------------------------:|:---------------:|:--------------------:|:-------------:|:-----------------------------:|:-------------:|:---------------------:|:-------------:|:---------------:|:-------------:|
> |                          |                 |         Siena        |    CHB-MIT    |          SleepEDF-20          |      SHHS     |          DEAP         |      SEED     |       Shu       |   BCI-IV-2a   |
> |          LaBraM          |        ×        |     0.9886±0.0043    | 0.9742±0.0099 |         0.7503±0.0388         | 0.7785±0.0243 |     0.5822±0.0321     |      OOM      |  0.5368±0.0312  | 0.2879±0.0160 |
> |  Single-task   EEGMamba  |        ×        |     0.9897±0.0053    | 0.9817±0.0036 |         0.8387±0.0399         | 0.8441±0.0163 |     0.5985±0.0247     | 0.5779±0.0584 |  0.6169±0.0467  | 0.4596±0.0547 |
> |         EEGMamba         |        √        |     0.9897±0.0038    | 0.9789±0.0132 |         0.8486±0.0276         | 0.8478±0.0177 |     0.5994±0.0134     | 0.5646±0.0366 |  0.6207±0.0505  | 0.4231±0.0522 |
>
> The difference between EEGMamba and these models is that:
> **1. End-to-End Training:** EEGMamba is trained end-to-end, allowing all components of the model to learn jointly, which makes EEGMamba highly efficient, as it avoids the need for multiple rounds of fine-tuning for each task. This is in contrast to models that train a shared backbone separately and then attach task-specific heads, which can be time-consuming and less efficient, especially when working with a large number of diverse tasks.
> **2. Multi-Task Learning Capability:** EEGMamba is designed to handle multiple tasks and datasets simultaneously within a single training session. Unlike existing foundation models, which can only process one dataset at a time during fine-tuning, EEGMamba is capable of delivering results from multiple datasets simultaneously in a single session.
> In summary, EEGMamba offers a more integrated and sophisticated approach to multi-task learning for EEG applications. We hope this clarification highlights the unique advantages of EEGMamba and addresses your query effectively.
>
> **Q2: How does the ST-Adaptive module capture interactions between channels and temporal features, which are critical in EEG data? Have you considered modeling these dependencies more explicitly?**
> **A2:** The ST-Adaptive module sequentially extracts the spatial features and temporal features of the EEG signals, without direct interactions. First, it uses a spatially adaptive convolution method to normalize EEG data to a fixed number of channels, enabling the model to capture channel interactions despite spatial diversity across datasets and tasks. Then the extracted channel features are input into the dual path structure in the tokenizer, which includes both small and wide kernel convolutions, allows the model to capture local and global temporal features. The resulting token naturally contains channel features and time features, and does not require additional spatio-temporal interaction.

---

> ### Author Response · Authors · 2024-11-23
> **Reply to Q3-Q5**
>
> **Q3: How would EEGMamba perform on out-of-distribution tasks or tasks not seen during training? Have you tested its ability to generalize to entirely new EEG task types?**
> **A3:** Thank you for the good suggestion. We use the Confused student EEG brainwave data [1] (hereinafter referred to as Confused EEG), which is a completely new task for EEGMamba. We applied the existing weights (i.e., the weights corresponding to Table 2-5 in the manuscript) to the Confused EEG, using 4 different random numbers to obtain a 7:3 training-test set ratio, similar to the approach in [3] (while [2] used five-fold cross-validation). We trained for 10 epochs each time and averaged the results. The test results are shown as follows:
> | Classification   Network | [2], 2019 | [3], 2023 | EEGMamba |
> |:------------------------:|:---------:|:---------:|:--------:|
> |    Reported   Accuracy   |   0.7500  |   0.7670  |    0.7825   |
>
> [1] Wang H, Li Y, Hu X, et al. Using EEG to Improve Massive Open Online Courses Feedback Interaction[C]//AIED workshops. 2013.
> [2] Wang H, Wu Z, Xing E P. Removing confounding factors associated weights in deep neural networks improves the prediction accuracy for healthcare applications[C]//Pacific Symposium on Biocomputing. Pacific Symposium on Biocomputing. NIH Public Access, 2019, 24: 54.
> [3] Lim Z Y, Neo Y L. Confused vs Non-Confused Electroencephalography Signal Classification Using Deep Learning Algorithm[C]//2023 IEEE International Conference on Automatic Control and Intelligent Systems (I2CACIS). IEEE, 2023: 195-200.
>
> **Q4: Could you clarify how this work builds on or differs from previous applications of SSMs or Mamba models in EEG classification? Adding this context could help readers better understand the specific contributions of EEGMamba.**
> **A4:** We searched for relevant keywords in Google Scholar and reviewed the literature published before the paper submission deadline. Most of the studies applying Mamba to EEG are still in preprint form and have not yet undergone peer review. Therefore, we provide a brief summary here to give readers an overview of the application of SSMs or Mamba models in EEG classification.
> [1] is the first application of Mamba to EEG used a single-directional Mamba model, but the article provides only a brief introduction to the methodology and experiments. [2] proposed a self-supervised learning (SSL) framework for sleep stage classification, utilizing a Mamba-based temporal context module to capture relationships among different EEG epochs, although the main component is still a Transformer-based model. [3] used Brain Timeseries Mamba and Brain Network Mamba module to encode the spatiotemporal features and long-term dependencies of EEG signals, respectively, to achieve efficient EEG classification.
> [1] Panchavati S, Arnold C, Speier W. Mentality: A Mamba-based Approach towards Foundation Models for EEG [J].
> [2] Lee C H, Kim H, Han H, et al. NeuroNet: A Novel Hybrid Self-Supervised Learning Framework for Sleep Stage Classification Using Single-Channel EEG[J]. arXiv preprint arXiv:2404.17585, 2024.
> [3] Behrouz A, Hashemi F. Brain-mamba: Encoding brain activity via selective state space models[C]//Conference on Health, Inference, and Learning. PMLR, 2024: 233-250.
>
> **Q5: Can you evaluate this on longer contexts to better get a sense for the necessity and usefulness for Mamba?**
> **A5:** Thank you for your suggestion. We can obtain the EEG signal length from each data set according to the rate×duration in Table 1. At present, for single-task EEGMamba, the maximum length is 128×60=7680 from the DEAP dataset, which we consider to be a very large length. When training with EEGMamba, our operation to unify the sampling frequency to 200Hz resulted in more datasets of larger length, such as the length of DEAP dataset is 200×60=12000, and the length of SHHS dataset is 200×30=6000. EEGMamba has the best performance among all models in this case.

---

### Meta-Review · Area_Chair_vczE · 2024-12-16

**Metareview:**

The work considers the problem of multitask decoding from EEG (to understand a model that can predict for different tasks without retraining). To so so it introduces a new architecture by using a bidirectional mamba state-space model and a MoE approach that can orient the model prediction based on a class token. Results are reported on 8 public datasets. To cope with the variability of input channels a different set of spatial filters are trained for each dataset.

Notable contributions:
- Mamba demonstrates memory efficiency and fast inference, making it advantageous for real-world applications.
- The model addresses the multi-task classification problem, showcasing how MoE enables prediction across a variety of downstream tasks.

Weaknesses:
- As pointed out by uVYA the motivation for multi-task training is not fully unclear. Effectively, the model cannot generalize to new datasets or subjects, particularly when new datasets vary in channel count.
- Comparison with SoTA including single tasks models are missing. While one can acknowledge the multi-task aspect of the work, as just pointed above taking a new dataset and a new task does effectively require retraining. There is no zero-shot transfer to new tasks here. So from a pure application point of view, if the objective is to have the best BCI model or the best sleep model it is not clear if the multitask approach is reasonable. Also as pointed out by uVYA results on very classical EEG BCI are pretty low. This questions the significance of the work.

While wapT is concerned about the claim of architectural novelty, the current decision is motivated by the 2 concerns above.

As a general comment to this community, it becomes very clear that benchmarks on such data need to be more standardized (test sets, validation strategy, etc.)

**Additional Comments On Reviewer Discussion:**

Reviewer wapT remains unconvinced after the discussion and acknowledged reading the feedback from the authors. Relevant concerns from uVYA are partially addressed (cf. key concerns in metareview). Feedback from xgg9 is weakly positive but rather shallow on the contribution. Overall no reviewer clearly champions this contribution, mostly due to unclear positioning of the work and no clear wins on a downstream task.

---

### Decision · Program_Chairs · 2025-01-22

Reject